# Hot-carrier trapping preserves high quantum yields but limits optical gain in InP-based quantum dots

Sander J. W. Vonk [1,2], P. Tim Prins [2], Tong Wang [3], Jan Matthys[4,5], Luca Giordano[4], Pieter Schiettecatte[4], Navendu Mondal [3], Jaco J. Geuchies [6,7], Arjan J. Houtepen [7], Jessi E. S. van der Hoeven [8], Thomas R. Hopper [3,9], Zeger Hens [4,5], Pieter Geiregat [4,5], Artem A. Bakulin [3] & Freddy T. Rabouw [1,2] ✉

Indium phosphide is the leading material for commercial applications of colloidal quantum dots. To date, however, the community has failed to achieve successful operation under strong excitation conditions, contrasting sharply with other materials. Here, we report unusual photophysics of state-of-the-art InP-based quantum dots, which makes them unattractive as a laser gain material despite a near-unity quantum yield. A combination of ensemble-based time-resolved spectroscopy over timescales from femtoseconds to microseconds and single-quantum-dot spectroscopy reveals ultrafast trapping of hot charge carriers. This process reduces the achievable population inversion and limits light amplification for lasing applications. However, it does not quench fluorescence. Instead, trapped carriers can recombine radiatively, leading to delayed—but bright—fluorescence. Single-quantum-dot experiments confirm the direct link between hot-carrier trapping and delayed fluorescence. Hot-carrier trapping thus explains why the latest generation of InP-based quantum dots struggle to support optical gain, although the quantum yield is near unity for low-intensity applications. Comparison with other popular quantum-dot materials—CdSe, Pb–halide perovskites, and $CuInS_2$—indicate that the hot-carrier dynamics observed are unique to InP.

Colloidal semiconductor quantum dots (QDs) have emerged as a promising candidate for many technological applications—such as lighting[1], solar cells[2], photon detectors[3], and optical computing[4]—owing to their facile processability and size-dependent optical properties. The development of high-quality QDs with tuneable visible emission started with CdSe[5,6], which contributed to the 2023 Nobel Prize in Chemistry. QD designs containing Cd or Pb remain those with the brightest emission and most precise control over properties[6–8].

[1]Soft Condensed Matter & Biophysics, Debye Institute for Nanomaterials Science, Utrecht University, Princetonplein 1, 3584 CC Utrecht, The Netherlands. [2]Inorganic Chemistry & Catalysis, Institute for Sustainable and Circular Chemistry, Universiteitsweg 99, 3584 CG Utrecht, The Netherlands. [3]Department of Chemistry and Centre for Processable Electronics, Imperial College London, W120BZ London, United Kingdom. [4]Department of Chemistry, Ghent University, Krijgslaan 281, 9000 Ghent, Belgium. [5]NOLIMITS, Core Facility for Non-Linear Microscopy and Spectroscopy, Ghent University, Krijgslaan 281, 9000 Ghent, Belgium. [6]Leiden Institute of Chemistry, Leiden University, Einsteinweg 55, 2333 CC Leiden, The Netherlands. [7]Opto-Electronic Materials Section, Faculty of Applied Sciences, Delft University of Technology, van der Maasweg 9, 2629 HZ Delft, The Netherlands. [8]Materials Chemistry & Catalysis, Debye Institute for Nanomaterials Science, Universiteitsweg 99, 3584 CG Utrecht, The Netherlands. [9]Department of Chemistry, University of Central Florida, 4111 Libra Dr, Orlando, FL 32816, USA. ✉e-mail: f.t.rabouw@uu.nl

However, as consumer applications demand QDs free of toxic elements, alternative materials have drawn considerable attention in recent years. InP-based QDs now offer photoluminescence efficiencies and a colour tunability that are on par with those of high-quality Cd- and Pb-containing designs[9–11]. This makes InP-based QDs an ideal phosphor for displays and lighting[12,13].

In stark contrast to these successes, InP-based QDs struggle in more demanding optical applications. In particular, the development of QD lasers using InP-based QDs lags behind other QD materials by more than two decades. Lasing from InP-based QDs has been reported only once[14], while the vast majority of the QD lasing literature has successfully used Cd-based or Pb–halide perovskite QDs[15–17]. Building on efforts starting in the early 2000s, QD lasers now show optically driven lasing—pulsed[18] and continuous wave[19]—and recently even electrically driven amplified spontaneous emission[20]. The near-total absence of InP-based QDs in the lasing literature is consistent with spectroscopic measurements, which have highlighted difficulties to achieve population inversion and gain from an ensemble of InP-based QDs[21].

In this Article, we combine ensemble and single-particle experiments to investigate why InP-based QDs are not yet suitable for lasing applications, despite their high brightness, and how their performance may be improved. On the ensemble level, we find a correlation between the magnitude of charge-carrier losses on the sub-ps timescales and slow delayed emission on the ns-to-μs timescales. Based on our observations, we propose that hot-carrier traps are most likely internal defects, for example located on the InP/ZnSe interface. This highlights the direction into which InP-based QDs should be improved for next-generation applications.

## Results

### Bright emitters without gain

We study InP/ZnSe/ZnS QDs[10] with a photoluminescence quantum yield of 95% for 420-nm excitation (Fig. 1a, high-resolution TEM image). Figure 1b shows the absorbance spectrum (black line) with the band-edge absorption at 2.13 eV. The PL peaks at 2.017 eV with a full width at half maximum (FWHM) of 168 meV (Fig. 1b red). Although this is clearly a bright and high-quality sample, our data shown below will evidence the involvement of charge-carrier traps in the excited-state decay.

Figure 1c shows a transient of the band-edge absorbance ($E_{probe} = 2.11$–$2.15$ eV) after intense 515-nm pulsed excitation (photon fluence $J = 8.2 \times 10^{15}$ cm$^{-2}$), exhibiting a characteristic shape found previously for InP-based QDs as well as other QD materials[21,22]. Three different stages of excited-state decay can be distinguished in the trace. In the cooling phase (within <1 ps after excitation, Fig. 1c blue), the absorbance decreases (bleaches) as hot carriers cool down to the band edge. Maximum bleach is observed when all hot carriers have cooled down to the band edge at around 1 ps (Fig. 1c, blue square), owing to (partial) blocking of the band-edge absorption transition. In the Auger phase (1–300 ps, Fig. 1c green), the absorbance recovers rapidly since multiexcitons decay non-radiatively via Auger recombination. We attribute the fitted time constants of 147 ps and 29 ps to the decay of biexcitons and higher multiexcitons, respectively (Suppl. Fig. 1)[21,23]. Finally, in the single-exciton phase (>300 ps, Fig. 1c red), the magnitude of the bleach (Fig. 1c red square) is determined by the remaining electron–hole pairs—at most one pair per QD—which recombine slowly (time constant 30 ns) leading to the gradual recovery of the absorbance.

The maximum bleach we observe in the experiment is weaker than expected (Fig. 1d). The band-edge absorbance bleach of our InP-based QDs saturates around $A \rightarrow 0$ as the excitation fluence is increased, while complete population inversion would lead to $A \rightarrow -A_0$ (where $A_0$ is the ground-state absorbance; Fig. 1e). Limited absorbance bleach in QDs was previously explained in terms of a high degeneracy of states at the valence-band edge[24] but this explanation does not reproduce the observed gain saturation at $A \rightarrow 0$ (Suppl.

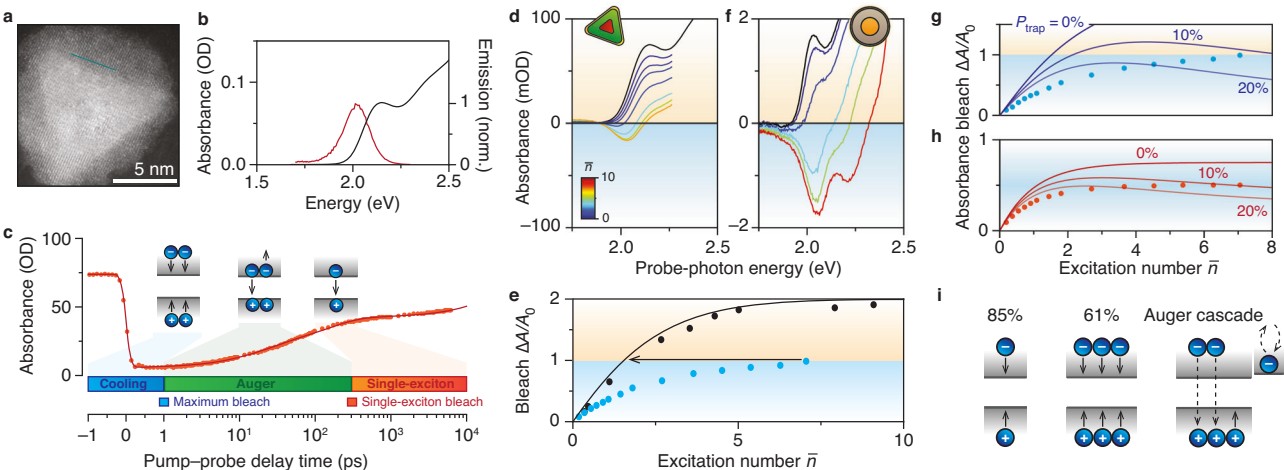

**Fig. 1 | Limited gain in InP-based quantum dots because of hot-carrier trapping. a** High-resolution TEM image of a InP/ZnSe/ZnS core–shell–shell QD. **b** Absorbance (black; left axis) and emission spectrum (red; right axis) spectra of the QDs. **c** Absorbance as a function of pump–probe delay time ($E_{probe} = 2.11 - 2.15$ eV, photon fluence $J = 8.2 \times 10^{15}$ cm$^{-2}$). In the cooling phase (blue) hot carriers relax to the band edge. The maximum bleach is reached at $t = 1$ ps after photoexcitation (blue square). In the Auger phase (highlighted in green) multiexcitons decay rapidly to single excitons. The single-exciton bleach level is reached at $t = 300$ ps (red square). The single-exciton phase (highlighted in red) shows slow radiative recombination of excitons. The solid line is a triple-exponential fit with time constants of 29 ps, 147 ps, and 30 ns. **d** Transient-absorbance spectrum after cooling ($t = 1$ ps) as a function of average number of excitations per pulse (differently colored lines; see color scale bar for the values of $\bar{n}$). Some gain develops at the low-energy side of the exciton transition. **e** The maximum bleach for InP/ZnSe/ZnS QDs (blue; $t = 1$ ps) and CdSe/CdS/ZnS QDs (black; $t = 1$ ps) as a function of the excitation number $\bar{n}$. See Supplementary Note 1 for our calculation of $\bar{n}$. Black line: prediction of a state-filling model for the maximum bleach, using electron and hole band-edge degeneracies $g_e = 2$ and $g_h = 4$. Arrow: the experiment on InP-based QDs at $\bar{n} = 7.1$ produces an initial bleach consistent with an effective number of excitations of no more than $\bar{n}_{eff} = 1.5$. **f** Same as **d**, but for CdSe-based QDs. Data reused from Ref. 25. **g** Maximum bleach as a function of the excitation number $\bar{n}$. Solid lines: predictions of our hot-carrier-trapping model (Supplementary Note 1) for different trapping probabilities $P_{trap}$. **h** Same as **g**, but for the single-exciton bleach ($t = 300$ ps). **i** The cooling probability $P_{cool}$ for $n = 1$ (85%) and $n = 3$ (61%) excitations for a hot-carrier trapping probability of $P_{trap} = 15$%. Source data are provided as a Source Data file.

Fig. 2)[21]. Instead, our data is consistent with the interception of photogenerated charge carriers on the timescale of cooling, a process commonly referred to as "hot-carrier trapping". These carriers never reach the band edge and do not contribute to the band-edge absorbance bleach nor gain. Historically, this problem occurred in CdSe-based QDs, but charge-carrier trapping could be prevented with improvements to the QDs' surface passivation, simultaneously leading to higher PLQYs of >80% and improved gain (Fig. 1f). These successes were consistent with the existing paradigm that PLQY and charge-carrier trapping are intimately linked. In contrast, in our InP-based QDs, the PLQY is high (95%) but the gain still appears limited by hot-carrier trapping. Below, we will obtain more direct evidence for hot-carrier trapping and find out why the PLQY is, nevertheless, near unity.

Comparing the experimental maximum bleach with a conventional state-filling model–assuming electron degeneracy $g_e = 2$ and hole degeneracy $g_h = 4$ equivalent to CdSe[25]–highlights that the hot-carrier losses are particularly severe for strong excitation (Fig. 1e). To construct this plot, we convert the photon fluence $J$ to an average number of excitations per QD per laser pulse $\bar{n} = \sigma J$ (Fig. 1e), where $\sigma$ is the absorption cross section at the excitation wavelength determined with two separate methods (Supplementary Note 1). We observe that $\bar{n} = 7.1$ excitations yield a bleach equivalent to no more than an effective number of $\bar{n}_{eff} = 1.5$ excitations (arrow in Fig. 1e). Clearly, the fraction of carriers lost during cooling increases for increasing $\bar{n}$.

Figure 1g, h show a model that reproduces the fluence-dependent hot-carrier losses observed experimentally (Supplementary Note 1). To achieve this, our model introduces a positive-feedback mechanism on hot-carrier losses: we assume that once a hot carrier is trapped, it acts as an efficient Auger acceptor that rapidly quenches all other excitations in the QD. Indeed, the assumption of rapid Auger processes involving localised excitations in QDs–even on the timescale of

cooling–is consistent with previous studies[26–28]. The simplest version of this model (Supplementary Note 1) qualitatively matches the experimental maximum bleach (Fig. 1g) and single-exciton bleach (Fig. 1h) for a hot-carrier trapping probability of $P_{trap} \approx 15\%$. The positive-feedback mechanism explains why trapping becomes increasingly problematic for increasing number of excitations. A single trapping event starts a cascade of Auger processes leading to significant losses (Fig. 1i). This decreases the overall probability of successful cooling from $P_{cool} = 1 - P_{trap} = 85\%$ for a single excitation ($n = 1$) to only $P_{cool} = (1 - P_{trap})^3 = 61\%$ for $n = 3$. Below, we will add more details to the model based on further experimental insights into the dynamics of hot carriers and achieve an improved match with the experiments (Supplementary Note 1).

## Measuring losses from hot-carrier states

To investigate hot-carrier trapping more directly, we perform pump–push–probe (PPP) spectroscopy experiments[24,29–33]. First, a pump laser at $E_{pump} = 3.1$ eV creates $\bar{n} = 0.1$ excitations per pulse (Fig. 2a, inset I). Subsequently, a sub-band-gap push pulse ($E_{push} = 0.52$ eV, 300 ps after the pump pulse) excites band-edge carriers to a hot-carrier state (Fig. 2a inset II). The absorbance changes induced by the push pulse are consistent with excitation of conduction-band electrons from the 1S to the 1P level (Suppl. Figs. 5, 6). In our experiments, a probe pulse measures the differential absorbance (difference between pump+push and push-only, see Methods and Suppl. Fig. 7 for details) at variable time delays with respect to the push. We minimise the push fluence to minimise the influence of multiphoton absorption (Supplementary Note 2). Exciting charge carriers by the push pulse lowers the band-edge bleach, which is restored as excited charge carriers cool down over a timescale of 770 fs. Importantly, only 92% of the initial bleach is recovered (Supplementary Note 2, corrected for the instrument-response function).

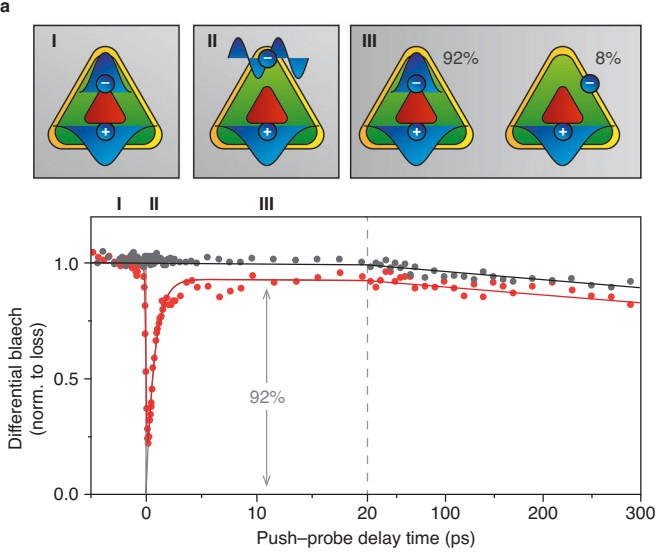

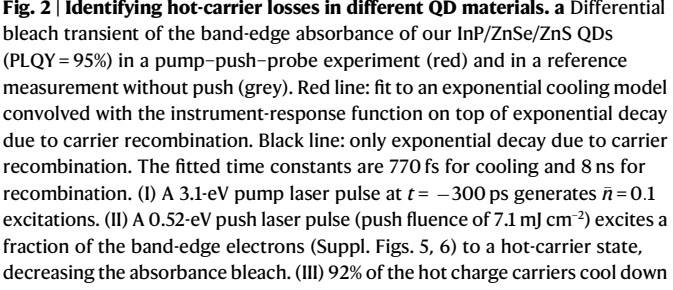

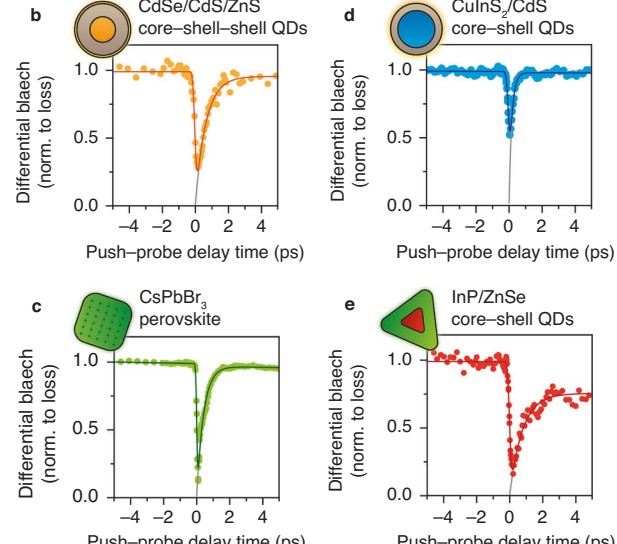

**Fig. 2 | Identifying hot-carrier losses in different QD materials. a** Differential bleach transient of the band-edge absorbance of our InP/ZnSe/ZnS QDs (PLQY = 95%) in a pump–push–probe experiment (red) and in a reference measurement without push (grey). Red line: fit to an exponential cooling model convolved with the instrument-response function on top of exponential decay due to carrier recombination. Black line: only exponential decay due to carrier recombination. The fitted time constants are 770 fs for cooling and 8 ns for recombination. (I) A 3.1-eV pump laser pulse at $t = -300$ ps generates $\bar{n} = 0.1$ excitations. (II) A 0.52-eV push laser pulse (push fluence of 7.1 mJ cm$^{-2}$) excites a fraction of the band-edge electrons (Suppl. Figs. 5, 6) to a hot-carrier state, decreasing the absorbance bleach. (III) 92% of the hot charge carriers cool down

to the band edge, but $P_{trap} = 7.5 \pm 0.8\%$ of the carriers are lost due to hot-carrier trapping, as measured at $E_{probe} = 2.07-2.20$ eV. Grey line: deconvolved PPP trace. The $y$-axis is normalised, setting the bleach just before the push to 1 and immediately after the push to 0. **b**–**e** Similar PPP experiments on (**b**) CdSe/CdS/ZnS QDs (PLQY = 95%), (**c**) CsPbBr$_3$ nanocrystals (estimated PLQY = 50–90%[7]), (**d**) CuInS$_2$/CdS QDs (estimated PLQY = 90%[35]), and (**e**) "bare" InP/ZnSe (estimated PLQY = 50%[34], $P_{trap} = 21.9 \pm 0.9\%$), without a ZnS shell. The fractional decrease of the absorbance bleach due to the push pulse is 37%, 49%, 62%, 19%, and 40% for the experiments in Fig. 2a–e, respectively. See Suppl. Fig. 11 for non-normalised data. Source data are provided as a Source Data file.

This means that $P_{trap} = 7.5 \pm 0.8\%$ (error provided as one standard deviation) of the hot electrons are lost (Fig. 2a, inset III). This evidence for hot-carrier trapping yields a trapping probability similar to the values we estimated from the pump–probe experiments (Fig. 1f, g, $P_{trap} = 15\%$). The discrepancy may indicate a dependence of $P_{trap}$ on the excess energy of the hot carriers, as confirmed by experiments with a higher push energy $E_{push} = 0.92$ eV yielding $P_{trap} = 10.5 \pm 0.6\%$ (Suppl. Fig. 9). Importantly, the hot-carrier trapping probabilities of $P_{trap} = 7.5\%$ at $E_{push} = 0.52$ eV and $P_{trap} = 10.5\%$ at $E_{push} = 0.52$ eV are higher than the fluorescence quenching probability under low-power illumination, which amounts to $1 - PLQY = 5\%$. This evidences that hot-carrier trapping does not necessarily lead to fluorescence quenching. Below we will investigate why.

PPP experiments on other state-of-the-art QD samples with a high PLQY indicate that significant hot-carrier trapping is a unique trait of InP-based QDs. CdSe/CdS/ZnS QDs (Fig. 2b), CsPbBr$_3$ perovskite nanocrystals (Fig. 2c), and CuInS$_2$/CdS QDs (Fig. 2d) all exhibit sub-ps cooling dynamics similar to InP, but the bleach recovers completely. In contrast, the hot-carrier losses are even higher (Fig. 2e, $P_{trap} = 21.9 \pm 0.9\%$) for InP/ZnSe QDs synthesized following the procedure introduced by Tessier et al.[34] This might highlight the importance of the additional interface oxidation treatment used for the InP/ZnSe/ZnS QDs used in this work[10], which possibly reduces defects responsible for hot-carrier trapping at the InP/ZnSe interface.

### Correlation between hot-carrier trapping and delayed emission

How is significant hot-carrier trapping consistent with the high quantum yield of 95% of our InP/ZnSe/ZnS QDs? To answer this question, we turn to room-temperature ensemble transient-photoluminescence (PL) measurements on the ns-to-μs timescale ($E_{laser} = 3.1$ eV, $\bar{n} \ll 1$). Figure 3a shows a transient-PL measurement of our sample around the emission maximum ($E_{det} = 2.02$–2.05 eV) on a double-logarithmic scale. Most of the emission (86%) can be described by a biexponential decay (fitted lifetime components of 13 and 41 ns), which we attribute to prompt band-edge recombination (Fig. 3b). Additionally, distributed power-law decay dynamics contribute 14% to the total emission (Suppl. Fig. 12). This distribution consists of decay rates spanning from significantly slower to rates approaching the prompt band-edge recombination. A similar power-law decay has been observed in various types of luminescent nanomaterials and termed "delayed emission"[35–37]. In this Article, we use the term "delayed emission" for any trap-related excited-state decay leading to slow emission compared to prompt band-edge recombination. This can be trap emission due to radiative free-to-bound recombination, in which a trapped and delocalised charge carrier recombine directly[38,39], or charge-carrier detrapping[36–38,40] restoring the emissive lowest-energy excitation. For example, both mechanisms are operative in CdSe nanoplatelets, which can be easily distinguished spectrally[38].

The approximate match between the hot-carrier-trapping probability ($P_{trap} = 7.5\%$ at $E_{push} = 0.52$ eV and $P_{trap} = 10.5\%$ at $E_{push} = 0.92$ eV) and fraction of delayed emission (Fig. 3a, 13.8 ± 0.7%) suggests a cause-and-effect relation, although this match alone is not enough to draw a solid conclusion. A tentative mechanism is schematically depicted in Fig. 3b: the excited-state decay pathway is determined by the choice between successful cooling to the band edge (probability $1 - P_{trap}$) or hot-carrier trapping (probability $P_{trap}$). Importantly, both pathways in Fig. 3b eventually lead to emission, so the PL quantum yield is unaffected by $P_{trap}$. This challenges the common association of trapping with decreased steady-state quantum yields[39,41,42]. The hypothesis of a causal link between hot-carrier trapping and delayed emission is strengthened by measurements on "bare" InP/ZnSe QDs, without an additional ZnS shell (Fig. 3c). Here, we observe a delayed-emission fraction as high as $35.5 \pm 1.5\%$, matching the high hot-carrier

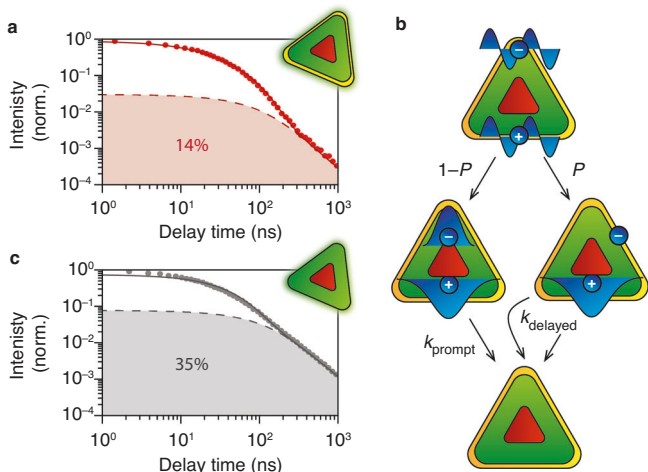

**Fig. 3 | Correlation between hot-carrier trapping and delayed emission.**
**a** Transient-photoluminescence (PL) measurements on InP/ZnSe/ZnS QDs on a double-logarithmic scale after nonresonant excitation ($E_{laser} = 3.1$ eV) and detection of the PL around the emission maximum ($E_{det} = 2.02 - 2.05$ eV). Solid line: model including prompt decay (biexponential) and delayed emission (power-law distributed). We find a relative contribution of delayed emission of $P = 13.8 \pm 0.7\%$, similar to the hot-carrier losses found in Figs. 1, 2. **b** Schematic of proposed excited-state decay pathways in InP/ZnSe/ZnS QDs. From a hot-exciton state, a fraction $P$ ($= P_{trap}$) of electrons is trapped and recombine with the hole on a time scale slower than band-edge recombination. **c** Same as **a**, but for bare InP/ZnSe QDs without a ZnS outer shell, which clearly shows more delayed emission ($P = 35.5 \pm 1.5\%$). Source data are provided as a Source Data file.

losses determined by PPP experiments (Fig. 2b, $P_{trap} = 21.9 \pm 0.9\%$). In the next section, single-particle measurements will turn out ideal to establish the cause-and-effect relation between hot-carrier trapping and delayed emission.

### Cause–effect relation between hot-carrier trapping and delayed trap emission

Below we will present single-particle experiments that provide direct evidence for the cause-and-effect relation between hot-carrier trapping (Figs. 1, 2) and delayed emission (Fig. 3). Moreover, we will find that direct trap emission, and not charge-carrier detrapping, leads to delayed emission in InP-based QDs. First, we deposit individual QDs on a glass coverslip, which we prepare and seal in a nitrogen-purged glove box to prevent air-induced bleaching. In our single-particle experiments, we split the QD emission between a spectrograph (50%) and a Hanbury-Brown–Twiss setup (50%) and confirm that the signal originates from a single QD from a cross-correlation analysis between the signals on the detection channels (Fig. 4a)[43].

Figure 4b shows a blinking trace of a single QD, which exhibits switching between a high-intensity ON state and one (or more) low-intensity OFF state(s). We focus our analysis on the ON state, selecting periods when the QD emits $I > 400$ cts / 100 ms and constructing the corresponding average-lifetime trace (Fig. 4c). Periods showing a longer lifetime (e.g. between $T = 2$–4 s in Fig. 4c) appear to be associated with lower-energy emission (Fig. 4d). Tentatively, we attribute this observation to fluctuations of $P_{trap}$ resulting in variations in trap-related emission.

To study the slow decay dynamics and associated emission spectra with a better signal-to-noise ratio, we construct a fluorescence-lifetime–intensity distribution focusing on the ON state (Fig. 4e). The lack of correlation between the emission intensity and excited-state lifetime shows that the PLQY is unaffected by changes in the lifetime. The histogram of lifetimes (Fig. 4e; top) is much broader than would be expected from noise due to sampling from a single exponential distribution (Fig. 4e, solid line, Suppl. Fig. 13). This shows that the excited-

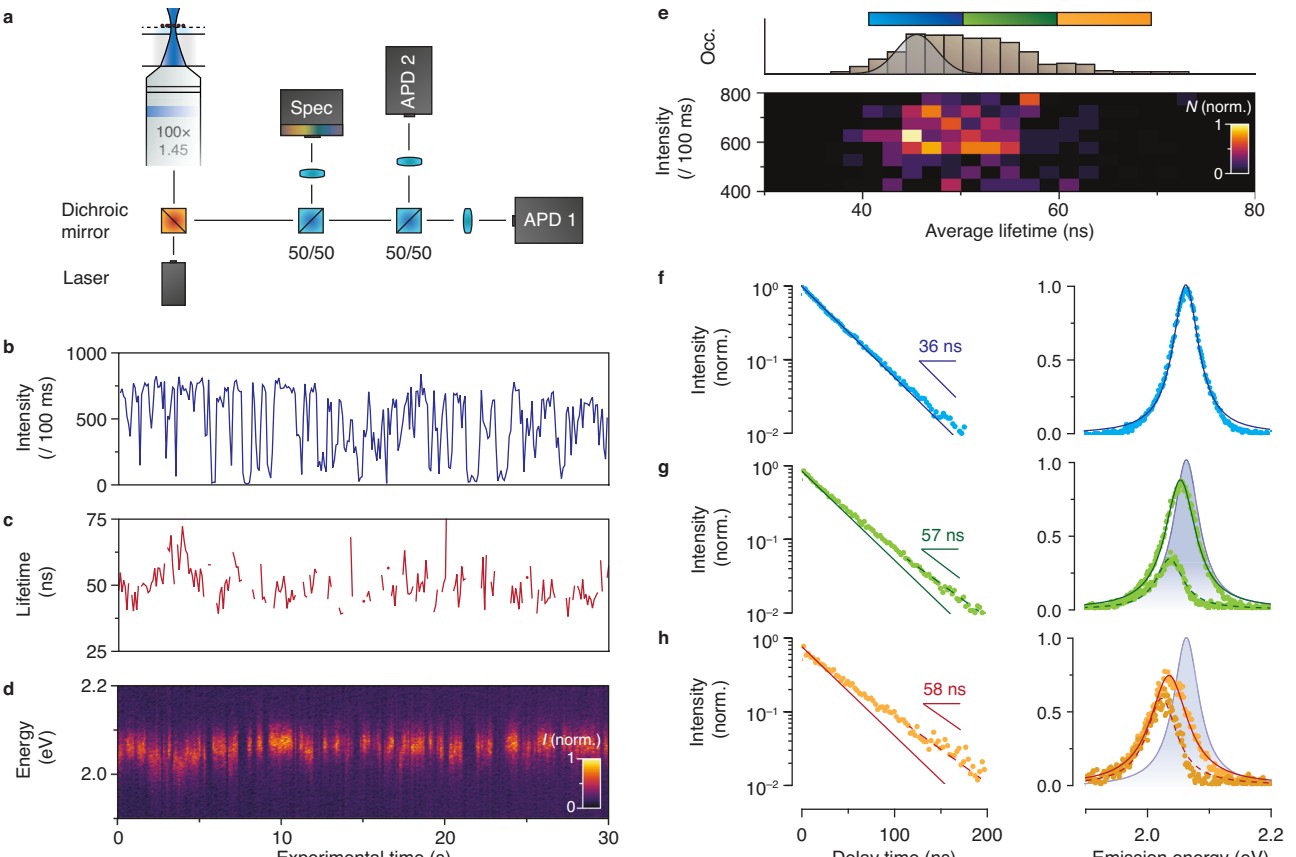

**Fig. 4 | Cause-and-effect relation between hot-carrier trapping and delayed trap emission. a** Setup for the single-QD experiments. The emission is split over a spectrograph (50%) and a Hanbury-Brown–Twiss setup (50%). **b** Blinking trace of a single QD showing switching between a high-intensity ON state and one (or more) low-intensity OFF state(s). **c** Average-lifetime trace of the ON state (count rate $I > 400$ cts / 100 ms). **d** Time trace of the emission spectra, showing a redshift of the emission spectrum correlated with a longer average lifetime (for example between 2–4 s). **e** Fluorescence-lifetime–intensity distribution of the ON state and the average-lifetime histogram. We subdivide the histogram into three regimes with short (blue, $\tau = 40$–50 ns), intermediate (green, $\tau = 50$–60 ns), and long (orange, $\tau = 60$–70 ns) excited-state lifetimes. **f** Transient-PL trace (left) and emission spectrum (right) for the short-lifetime regime (blue range in panel **e**). The PL decay is a single exponential with a fitted lifetime of 35 ns. The emission spectrum has an emission maximum at 2.064 eV and a FWHM of 51 meV. **g** Same as **f**, but for the

intermediate-lifetime regime (green range in panel **e**). The transient-PL trace is a biexponential with lifetime components of 35 ns (solid line; assigned to band-edge emission, 65% of total) and 57 ns (dashed line; assigned to trap-state emission, 35% of total). The total emission spectrum is redshifted by 9.5 meV with respect to panel **f** (blue shaded spectrum) and is broader with a FWHM of 60 meV. Dashed line: trap-state emission spectrum, peaking at 2.037 eV, reconstructed by subtracting the band-edge emission contribution from the total emission spectrum. **h** Same as **f** and **g**, but for the long-lifetime regime (orange range in panel **e**). The contribution of 35-ns band-edge emission and 58-ns trap-state emission are 35% and 65%, respectively. The total emission spectrum is redshifted by 30 meV with respect to panel **f** (blue shaded spectrum) and is even broader with a FWHM of 73 meV. The reconstructed trap-state emission spectrum peaks at 2.023 eV with a FWHM of 60 meV. Source data are provided as a Source Data file.

state lifetime of our QD indeed changes over the course of the experiment. We use the lifetime histogram to select moments in our experiments where the QD exhibits a short (blue, $\tau = 40$–50 ns), intermediate (green, $\tau = 50$–60 ns), or long (orange, $\tau = 60$–70 ns) excited-state lifetime. When the lifetime is short, we observe single-exponential decay with a fitted lifetime of 35 ns (Fig. 4f, solid blue line), which we attribute to band-edge emission. The short-lifetime emission spectrum (Fig. 4f, right) is centred around 2.064 eV and has a FWHM of 51 meV.

The decay dynamics and emission spectra during moments of longer average lifetime provide proof for a causal link between hot-carrier trapping and delayed trap emission. During selected moments with an intermediate average lifetime (Fig. 4g), we observe biexponential PL decay. The short component has a lifetime matching the band-edge emission (35 ns, 65% of total emission, solid green line) but with a reduced amplitude (compare Fig. 4f, g). Single-QD PL decay with a lifetime equal to band-edge emission, but a lower amplitude, is evidence for hot-carrier trapping, and has been used to identify "B-type blinking" in Ref. 44. In our InP-based QDs, the reduced amplitude is

accompanied by the appearance of a relatively slow decay component (Fig. 4g right, fitted lifetime of 57 ns, 35% of total emission). Simultaneously, the total emission spectrum (Fig. 4g) is redshifted by 9.5 meV and broader (FWHM = 60 meV) than the band-edge emission spectrum in Fig. 4f. This combination of observations is consistent with hot-carrier trapping leading to trap emission. Alternative explanations, based on the quantum-confined Stark effect or charge-carrier detrapping before recombination, are inconsistent with the data (Suppl. Fig. 14). During selected moments with the longest average lifetime (Fig. 4h), the PL decay amplitude is even smaller, and the total emission spectrum is more redshifted. We conclude that the hot-carrier-trapping probability $P_{trap}$ fluctuates on the timescale of the experiment. The PL decay dynamics show varying trapping probabilities between $P_{trap} = 0\%$ (Fig. 4f) and $P_{trap} = 65\%$ (Fig. 4h). Indeed, we obtain approximately the same trap-emission spectrum from the emission spectra of Fig. 4g, h if we subtract band-edge contributions based on the values of $P_{trap}$. The delayed-emission lifetime of 57–58 ns contributes to the fast end of the power-law decay-rate distribution observed on the ensemble scale (Fig. 3a). Indeed, other single QDs

from the same synthesis batch show varying delayed-emission lifetimes (Suppl. Fig. 15).

Fluctuations in excited-state dynamics on the single-QD level are not unique to InP-based QDs. However, the key observation proving hot-carrier trapping and delayed trap emission—intermittent biexponential decay with a lower amplitude and a slow component (Fig. 4g, h)—has not yet been reported for other QD materials. To confirm that InP-based QDs are different, we re-analysed the fluctuating excited-state lifetimes of $CuInS_2$ and CdSe single-QD measurements of Hinterding et al.[45,46] and performed additional measurements on a Pb−halide perovskite-based sample. The lifetime fluctuations in these materials are qualitatively different from InP-based QDs, and inconsistent with delayed emission following hot-carrier trapping (Supplementary Note 3).

The characteristics of the trap emission are most consistent with hot-carrier trapping due to internal defects, for example at the InP/ZnSe core−shell interface. Trap emission from surface traps is typically strongly broadened (linewidths of 0.3 eV) and redshifted (by up to 0.6 eV) compared to band-edge emission[47]. In contrast, the trap emission from our InP-based QDs has a relatively narrow linewidth and minor redshift compared to the band edge (Fig. 4)[47]. The redshift and broadening of trap emission are due to vibrational coupling, which is expected to be weaker for an internal trap compared to a surface trap (Suppl. Fig. 20). Indeed, the interior of a QD is simply more rigid than the outer surface[48]. Moreover, Fröhlich interaction with polar phonons depends on the spatial charge separation, which is smaller when a charge is trapped internally compared to one trapped on the QD surface[49]. The minor broadening and redshift of our trap emission is thus indicative of internal trapping. Comparison of the trap emission intensity (Suppl. Fig. 19) for different excitation photon energies confirms this: the trapping probability hardly depends on the excitation energy, while a surface trap would be more accessible upon excitation at energies beyond the shell bandgap.

## Discussion

While the limited gain of InP-based QDs (Fig. 1) was already indicative of hot-carrier trapping, the experiments in Figs. 2−4 provided (direct) evidence and more insights into the process. In particular, the delayed-emission measurements (Figs. 3, 4) show that the state with a trapped carrier has a finite oscillator strength, smaller than regular band-edge recombination by a factor of order 1–10. An adapted hot-carrier-trapping model (Supplementary Note 1), including a finite oscillator strength and a fraction of QDs without trapping, provides an improved match to the absorbance bleach data of Fig. 1g, h.

Encouragingly, the transient-absorbance spectra of Fig. 1 show a small redshifted gain feature, due to the Stokes shift between absorption and emission[21]. Unfortunately, the gain saturates at no more than $-0.25A_0$, while hypothetical QDs without losses would reach $-A_0$. Our adapted hot-carrier trapping model reproduces the gain spectrum directly after cooling (Fig. 5a, c) and the maximum optical gain (Fig. 5d) as a function of $\bar{n}$, if we include a Stokes shift. In line with these observations, time-resolved gain spectra reveal that the gain redshifts over the course of the recombination Auger phase (0–200 ps) as the charge carrier population decreases (Fig. 5b and Suppl. Fig. 17). While a Stokes shift is useful for achieving gain, further analysis explains why Stokes-shifted gain is even more sensitive to hot-carrier trapping than the bleach at the absorption maximum (Suppl. Fig. 18).

The trapping dynamics in InP-based QDs explain their contrasting performance in displays compared to lasers. As trapping occurs on ultrafast timescales (Fig. 2), optical gain is negatively affected even directly after a laser pulse (Fig. 1). This is problematic for laser applications. A clever solution to circumvent hot-carrier trapping may be resonant pumping of the QDs. However, one would then operate the QD as a 2-level system and, unfortunately, the fundamental laws of stimulated emission make population inversion by optical pumping of

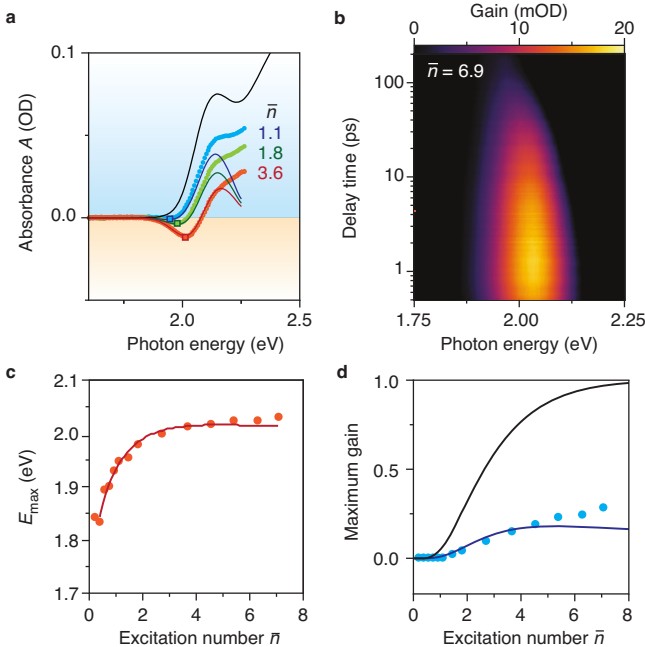

**Fig. 5 | Early saturation of thresholdless gain for InP-based QDs. a** Excited-state absorbance spectra at $\bar{n} = 1.1$ (blue), $\bar{n} = 1.8$ (green), and $\bar{n} = 3.6$ (red), after the cooling phase, corrected for the redshifted induced-absorption feature. Solid lines: excited-state absorbance spectra from our simple model including hot-carrier trapping with $\bar{P}_{trap} = 27\%$, a 30% relative oscillator strength of the trap state, and a Stokes shift between stimulated emission and absorption of $\Delta E = 70$ meV. **b** Time-resolved gain spectra (masked excited-state absorption $A < 0$) for $\bar{n} = 7.1$. We observe Stokes-shifted gain up to 200 ps after photoexcitation and the photon energy of maximum gain slightly redshifts with time. **c** The maximum-gain energy $E_{max}$ blueshifts with $\bar{n}$. Solid line: numerically found $E_{max}$ versus $\bar{n}$ using our model including hot-carrier trapping (Supplementary Note 1). **d** Experimental maximum optical gain $-A/A_0$ (blue dots) as a function of the excitation number $\bar{n}$. We observe that our InP-based QDs show thresholdless gain, which saturates at 25% of the maximum possible gain. Solid blue line: numerically found maximum optical gain using our model including hot-carrier trapping (Supplementary Note 1) Black solid line: numerically found maximum optical gain for regular state filling without hot-carrier trapping, which saturates at complete inversion of band-edge absorption ($-A/A_0 = 1$). Source data are provided as a Source Data file.

a 2-level system impossible[50]. The possibility of nonresonant optical pumping is exactly what makes QDs so interesting for lasing.

Our results highlight a key challenge in making InP-based QDs ready for lasing application: the suppression of hot-carrier trapping. Until now, InP-based QDs have been optimised mostly for brightness in display applications, so there has been no need to prevent the type of hot-carrier dynamics uncovered in this work. This explains why the problem of hot-carrier trapping can persist even in the latest generation of bright InP-based QDs. As the trap emission observed is most consistent with internal traps, the internal structure of the QDs seems the most important target. A potential avenue lies in wave-function engineering by interface polarisation[51]. InP cores can form polarised bonds with the ZnSe shell material that yield inward or outward-pointing dipole moments depending on the InP surface termination. Outward dipole moments could distance the electron wave function from potential interfacial traps on the core−shell interface and prevent hot-carrier trapping. Other strategies could attempt to circumvent the formation of internal and interfacial trap states all together, for example by using MgSe[52] shelling, which lowers the lattice mismatch between the core and shell material. The difference in hot-carrier trapping probability between the sample of Figs. 2a, 3a and the sample of Figs. 2e, 3c shows a beneficial effect of an oxidation treatment of the core surface during the synthesis[10,34]. The challenge for InP-based QDs

is clearly different from the challenge faced by CdSe-based QDs two decades ago[53]. The lasing performance of CdSe-based QDs has greatly improved over the past two decades mostly owing to core–shell designs that suppress Auger decay. While such steps may also be necessary for InP-based QDs, especially for CW lasing, these may not be sufficient. Indeed, hot-carrier losses reduce gain already before the Auger phase. For a more rational design of the next generation of InP-based QDs, density-functional theory may be used to explore hot-carrier trap states inside the bands, and not just inside the bandgap. In view of the significant Stokes shift, InP-based QDs hold great potential as a laser gain medium, as soon as the unfavourable hot-carrier dynamics are under control.

## Methods

### Synthesis procedure InP/ZnSe/ZnS quantum dots

Colloidal InP/ZnSe/ZnS were synthesized using the synthesis procedures as reported in Ref. 10. The synthesis was started by adding $InCl_3$ (0.45 mmol, 99.999%, Sigma-Aldrich) and $ZnCl_2$ (2.20 mmol, ≥98%, Sigma-Aldrich) to 3 mL of anhydrous oleylamine (80–90%, Acros organics). Subsequently, the mixture was stirred and degassed at 120 °C for 1 h and then heated to 180 °C under inert atmosphere. Upon reaching 180 °C, tris(diethylamino)phosphine (1.83 mmol, 97%, Sigma-Aldrich)–transaminated with 2 mL of anhydrous oleylamine–was injected in the reaction mixture. After 30 min, the dispersion was cooled to 120 °C, and tetrabutylammonium hexafluorophosphate (0.31 mmol, 98%, VWR), 0.3 mL of water, and of zinc(II) oleate (3.18 mmol) mixed in 2 mL of anhydrous oleylamine and 4 mL of anhydrous 1-octadecene (90%, Alfa Aesar) were added sequentially. Subsequently, the mixture was stirred and degassed for 1 hour. Next, 1.6 mL of a stoichiometric TOP-Se (2.24 M) solution was injected, and the temperature was raised to 330 °C. At this temperature, the shell growth proceeded for 50 minutes. The reaction mixture was cooled down to 120 °C, after which of zinc(II) acetate (2.21 mmol) was added and the mixture was stirred and degassed for 1 h. Consecutively, 1 mL of a stoichiometric TOP-S (2.24 M) solution was injected, and the temperature was raised to 300 °C. After 1 h of ZnS shell growth, the temperature had been set to 240 °C and 1 mL of 1-dodecanethiol (≥98%, Sigma-Aldrich) was injected. Ten minutes later, the reaction was stopped by cooling the mixture down to room temperature. InP/ZnSe/ZnS QDs were then precipitated once using anhydrous ethanol, redispersed in anhydrous toluene, and stored in a $N_2$-filled glovebox.

### High-resolution transmission-electron microscopy

High resolution electron microscopy was performed using a double aberration corrected Spectra 300 (Thermo Fisher Scientific). The high-angle annular dark-field scanning transmission electron microscopy (HAADF-STEM) images were recorded with 1024 × 1024 pixels, 13.7 pm pixel size, a dwell time per pixel of 10 μs, a magnification of 7.16 Mx and spot size 9. The sample was prepared by drop casting the QD dispersion on a TEM grid and removal of the organic ligands through washing with activated carbon[54]. In a small beaker, activated carbon was mixed with ethanol. The grid was submerged in this dispersion for 10 minutes and then left to dry.

### Transient-absorption measurements

**PP experiments.** The sample was excited using 190-fs pump pulses at 2.41 eV, which were generated by frequency doubling 1030-nm pulses emitted from a pulsed Yb:KGW laser operating at a frequency of 2.5 kHz (Light Conversion, PHAROS). The probe pulses were generated by focusing a small portion of the 1030-nm pulses into a 10-mm thick YAG crystal (EKSMA Optics), creating a supercontinuum ranging from 550 to 950 nm. These probe pulses were delayed relative to the pump pulses using a delay stage with a maximum delay of 6 ns. For the measurements the sample was dispersed in an optically

inactive solvent (toluene) in a 2 mm thick cuvette, to achieve an absorbance of approximately 100 mOD at the wavelength of the pump, which provided a balance between signal strength and an approximately uniform pump intensity throughout the cuvette. During the measurements, the sample was continuously stirred to avoid charging or photodegradation, and care was taken to avoid exposure to ambient conditions.

**PPP experiments.** The PPP experiments were based on a commercial Helios transient absorption spectroscopy setup (Spectra Physics, Newport Corp.). Ultrafast 800 nm (1 kHz, <100 fs pulse duration) laser pulses were generated by a Ti:sapphire regenerative amplifier (Solstice, Spectra Physics, Newport Corp.) and split into three portions. The first portion was directed to a beta barium borate (BBO) crystal for the 400 nm pump pulse via second-harmonic generation. The second portion was sent into an optical parametric amplifier (TOPAS Prime, Light Conversion) and a frequency mixer (NIRUVIS, Light Conversion) for the NIR push pulse. The remaining portion of the fundamental 800 nm light was focused into a sapphire crystal to generate the broadband white light probe in the visible region. In the setup, the path length for the push beam was fixed, and the pump and probe were delayed by separate mechanical stages. The three beams were focused onto the sample at the same spot (diameter ~0.5 mm), and the transmitted probe was collected by a CMOS fibre spectrometer. To mitigate shot-to-shot noise, we used another fibre spectrometer to record the fluctuations in a reference beam split off from the probe by a neutral density filter prior to hitting the sample. A mechanical chopper in the pump beam was modulated at 500 Hz to block every other pulse and relay the TA signal. The colloidal QD solutions were sealed in 5 mm-thick quartz cuvettes (110-5-40, Hellma Analytics, Germany) inside a nitrogen-purged glovebox, and stirred continuously by a magnetic stirrer bar during laser measurements.

### Single-QD measurements

All single-QD measurements were performed on a home-built optical setup consisting of a Nikon Ti-U inverted microscope body. The single-QD samples were prepared by dropcasting 10 μL of a diluted ($10^4$ dilution factor) solution of QDs onto a glass coverslip inside a nitrogen-purged glovebox. The sample was sealed by a spacer between the coverslip and a microscope slide to ensure an oxygen-free environment during the measurements. On the microscope, a 405-nm pulsed laser (PicoQuant D-C 405, controlled by PicoQuant PDL 800-D laser driver), operated at 1 MHz repetition rate, was guided to the sample by a dichroic mirror (edge at 425 nm, Thorlabs DMLP425R) and focused by an oil-immersion objective (Nikon CFI Plan Apochromat Lambda 100× NA 1.45) onto the sample. The QD emission was collected by the same objective and half of the emission was guided to two single-photon detectors in a Hanbury-Brown–Twiss setup. The emission was split by a non-polarising beamsplitter (Thorlabs BS013) and focused (achromatic aspherized lens Edmund optics, 49-659) onto a single-photon avalanche diode (SPAD); Micro Photonic Devices PDM, low dark counts <5 Hz). The other half of the emission was guided to a spectrometer (Andor Kymera 193i, 150 lines/mm reflective grating) with an electron-multiplying CCD detector (Andor iXon Ultra 888). The function generator, both SPADs, and the laser driver were connected to a quTools quTAG time-to-digital converter, which communicated all photon detection events and laser pulses to a computer. Home-written software was used for live data visualisation (e.g. photon-correlation function) and data storage.

## Data availability

Source data are provided with this paper. They have also been deposited in and are available from the Zenodo repository with https://doi.org/10.5281/zenodo.15544826[55]. Source data are provided with this paper.

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

## Acknowledgements

We thank Maksym Kovalenko and Maryna Bodnarchuk for the Pb–halide-based perovskite reference sample. S.J.W.V. and F.T.R. acknowledge support from the Dutch Research Council NWO (OCENW.KLEIN.008 and Vi.Vidi.203.031), and by The Netherlands Center for Multiscale Catalytic Energy Conversion (MCEC), an NWO Gravitation Programme funded by the Ministry of Education, Culture and Science of the Government of The Netherlands. T.W., N.M., and A.A.B. acknowledge support from the European Research Council (ERC) under the European Union's Horizon 2020 research and innovation program (Grant Agreement 639750/VIBCONTROL) and from UKRI/EPSRC (ActionSpec, Grant Ref: EP/X030822/1). N.M. and A.A.B. acknowledge support from the European Commission through the Marie Skłodowska-Curie Actions under Horizon 2020 (Project PeroVIB, 101018002). T.R.H. acknowledges support from an EPSRC Doctoral Prize Fellowship. J.M. and P.G. acknowledge ERC Starting Grant 'NOMISS' (Grant no. 101077526) and UGent Core Facility Program (NOLIMITS) for funding. P.S., L.G., and Z.H. thank FWO-Vlaanderen (12A9123N) for research funding. For access to the TFS Spectra300 microscope at EM Utrecht, we acknowledge the Netherlands Electron Microscopy Infrastructure (NEMI), project number 184.034.014, part of the National Roadmap and financed by the Dutch Research Council (NWO).

## Author contributions

L.G., P.S., and J.J.G. synthesized particles under the supervision of Z.H. and A.J.H. J.E.S.v.d.H. performed electron microscopy on the particles. T.W., J.M., N.M., and J.J.G. performed transient absorption measurements under the supervision of A.J.H., T.R.H., Z.H., P.G., and A.A.B. S.J.W.V., P.T.P., T.W., J.M., N.M., and J.J.G. analyzed the transient absorption measurements under the supervision of A.J.H., T.R.H., Z.H., P.G., A.A.B, and F.T.R. S.J.W.V. performed and analyzed single-particle experiments on the particles under the supervision of F.T.R. All authors contributed to the interpretation of the data. S.J.W.V. and F.T.R. wrote the manuscript with input from all authors.

## Competing interests

The authors declare no competing interests.
