## [Peer Review File · Nature Communications]

Hot-Carrier Trapping Preserves High Quantum Yields but Limits Optical Gain in InP-Based Quantum Dots

Corresponding Author: Dr Freddy Rabouw

Version 0:

Reviewer comments:

Reviewer #1

(Remarks to the Author)

This study delves into the limitations of InP QDs, particularly focusing on hot carrier trapping and delayed fluorescence dynamics. While the investigation of hot carrier trapping via pump-probe-push spectroscopy is intriguing, the correlation drawn between delayed fluorescence and trapped hot carriers lacks substantial evidence. The authors attribute three time constants extracted from transient absorption results to the decay of various excitonic states. However, this assertion lacks robust support and contradicts previous discussions regarding the timing of the Auger phase.

The discussion on the dynamic behavior of the SE peak in the transient absorption spectrum would benefit from the inclusion of the actual spectrum itself. Without visual aid, the clarity of the analysis is compromised, leaving readers to speculate on the nature of the observed dynamics.

Fig. 2 b-e presents data on the photoluminescence quantum yields (PLQYs) of reference QDs with different compositions. However, the text fails to provide these values, leaving readers without crucial information necessary for understanding the significance of the presented results.

The authors posit that despite the high probability of hot carrier trapping in InP QDs, the achievement of high PLQY is attributed to "delayed fluorescence." However, the assertion that the probability of hot carrier trapping is comparable to the amplitude of delayed fluorescence lacks substantiation. Furthermore, there is a conspicuous absence of explanation regarding the distinction between "delayed fluorescence" and conventional trap emission, as well as how this phenomenon contributes to the enhancement of PLQY in the QDs. Clarification on these points is essential for a comprehensive understanding of the proposed mechanism.

Reviewer #2

(Remarks to the Author)

The manuscript entitled "Hot-Carrier Trapping Preserves High Quantum Yields but Limits Optical Gain in InP-Based Quantum Dots" by Freddy Rabouw et al. investigates the photophysics of Indium Phosphide (InP)-based quantum dots (QDs), focusing on the phenomenon of hot-carrier trapping. They claim that the trapping maintains high quantum yields but limits optical gain, making these QDs unsuitable for high-intensity applications like lasers. The paper presents comprehensive evidence for the idea of hot-carrier trapping, combining both ensemble-based and single-quantum-dot spectroscopy to understand the limitations of InP-based QDs. They also used advanced spectroscopic techniques, such as time-resolved spectroscopy and pump-push-probe experiments, to provide robust and detailed insights into the hot-carrier dynamics. The comparison with other quantum dot materials effectively highlights the unique challenges associated with InP-based QDs. However, some experimental results need to be further validated, especially the pump-push-probe part, which plays a key role in the whole paper. For these reasons, I recommend a major revision before considering this work for publication. My concerns are as follows:

One of my concerns is about the pump-push-probe results, which provide direct evidence of hot-carrier trapping in the manuscript. The pump-push-probe is a complicated experimental technique that involves three-pulse interaction. In this paper, a near-infrared pulse (0.95 eV, 1.3 μm) was used as a push to excite the cold carrier to the hot carrier population. The cold carrier absorption of photons requires assistance from phonons, making the absorption cross-section very low. For this

reason, the push pulse always requires high fluence to observe hot-carrier excitation. Under high push fluence, the quantum dot system is very likely to have interband gap transitions due to multiphoton absorption. When there is interband transition induced by the push pulse, it will result in a reduced bleach signal, which will be similar to the observed “hot-carrier trap.” The authors also reported different push pulse energies (0.51 eV) observing different results, which could also be due to the lower multiphoton absorption cross-section at lower energy. To exclude this possibility, I recommend providing more details of experiments such as push fluence, and conducting multiphoton absorption experiments for all the samples to confirm the multiphoton absorption threshold.

The paper describes delayed emission attributed to hot-carrier trapping. Can the authors provide more insights into the nature of the trap states? The authors claim that hot-carrier trapping limits optical gain. Will resonant excitation induce lasing in this material? Additionally, the paper does not sufficiently explore potential methods to mitigate hot-carrier trapping in InP-based QDs, which limits the practical applicability of the findings. Can the authors propose specific synthesis modifications or experimental techniques that might mitigate hot-carrier trapping?

Some experimental details, such as the exact conditions of the spectroscopic measurements (e.g., transient absorption spectrum, pump-push-probe excitation fluence) and fitting parameter uncertainty, are not fully elaborated, which might hinder reproducibility.

Overall, while the study provides significant insights into the limitations of InP-based QDs, addressing these concerns would greatly strengthen the manuscript and its contributions to the field.

Reviewer #3

(Remarks to the Author)

The authors basically report on the importance of Auger processes in the relaxation and recombination scenario of photoexcited charge carriers. Such processes have been already investigated in InP-based quantum dots as partly mentioned in the text. It is known that strong Auger processes hinder lasing processes.

Version 1:

Reviewer comments:

Reviewer #1

(Remarks to the Author)

The authors have revised the manuscript according to the comments of reviewers and most issues have been well addressed. One additional recommendation would be to keep the Figure Captions more concise as most of the current Figure Captions are much longer than the paragraphs in the main text and a good portion of them are redundant with the main text. Lengthy captions hinder the readers from quickly identifying what is presented in the following figure while reading the paper.

Reviewer #2

(Remarks to the Author)

I still have concerns about the pump-push-probe setup based on the authors' reply. They state that:

“...the pump–push–probe signal, should not directly contain push–probe contributions such as multiphoton absorption... Furthermore, we note that the first-order effect of multiphoton absorption would be an increased bleach due to the push pulse, because of additional band-edge charge carriers...”

This is inaccurate. The differential absorbance, which is the signal comparison between $A_{\text{pump on, push on, probe on}}$ and $A_{\text{pump off, push on, probe on}}$ measures the effect of the pump pulse. When the push pulse induces multiphoton absorption (including two-photon or three-photon absorption), some carriers are excited to the conduction band. As a result, the difference signal caused by the pump pulse is weakened because fewer carriers are available for excitation. Therefore, the push pulse would lead to a reduced bleach signal, not an increase.

The authors also mention:

“Nevertheless, we characterised the multiphoton absorption of our semiconductor QDs in more detail. We performed pump–push–probe experiments at increasing push fluence and with different push energies. The results are shown below (Supplementary Fig. S6). The apparent hot-carrier losses are constant at low fluence below 10 mJ cm^{-2} but increase for higher push fluences. The increased hot carrier losses for higher fluences must be an indirect effect of interband multiphoton absorption, because the direct effect of interband absorption is the generation (rather than loss) of carriers. Indeed, control experiments with the pump pulse blocked show significant three-photon-induced bleach at push powers exceeding 10 mJ cm^{-2} .”

While the power-dependent experiments provide useful insight into push pulse effects, it should also be noted that push-pulse-induced two-photon absorption follows a quadratic relationship, suggest that the initial start would be very flat. From their push-probe experiments, at 4.6 mJ/cm^2 , multiphoton absorption should already be occurring. Based on their data, I cannot conclude that the observed carrier losses are due to hot carrier trapping.

Version 2:

Reviewer comments:

Reviewer #2

(Remarks to the Author)

The revised manuscript has significantly improved the analysis of the pump-push-probe experiments. The authors conducted push power-dependent experiments to evaluate the relative contributions of hot-carrier losses and multiphoton absorption. They argue that at low push fluences, the influence of multiphoton absorption is minimal, and the constant P_{trap} represents the real hot-carrier trapping probability. However, one critical aspect that the authors have overlooked is the role of carrier recombination during the hot-carrier cooling process. Given that carrier recombination is not particularly slow for these materials, the extracted bleach signal from fitting should not remain constant, as recombination effects would influence the signal. Although the analysis appears reasonable, the data support remains weak, and I still question the manuscript's conclusion that the observed results from pump-push-probe (PPP) experiments are mainly due to hot-carrier losses rather than multiphoton effects. No further review is required; I will leave the decision to the editor.

Reviewer #4

(Remarks to the Author)

I understand from reading the material sent that my part was to decide between two conflicting views of the potential artifacts which might arise from multi-photon absorbance of the Push pulse on the PPP data. In order to calibrate my response I first read the entire article and find that significant misconceptions have been incorporated into the analysis which make publication of this paper premature.

The assumption that absence of a full reversal of band edge absorption to stimulated emission after absorbing multiple photons is due to trapping goes against prevailing opinion in the field. Starting with the earliest power dependent studies of CdSe nano-crystals by Klimov and Co, the induced bleach saturated at sample transparency, with no sign of stimulated emission. Furthermore a single exciton was shown to induce a 50% reduction in the lowest exciton absorbance. Ironically, the mechanism behind this is still debated, but is often assigned to degeneracy of the hole states which reduce their contributions to state filling. Since the authors assume similarity in electronic states between CdSe and InP, it stands to reason that a similar explanation applies here as well. The above assumption is not a limited issue and is at the heart of all analysis here. Therefore not dealing with this issue is a major shortcoming which requires further thought.

Some additional points that I suggest the authors address are:

The extensively normalized format of the PPP presentation is problematic. What is the fraction of the bleach which is erased due to the push and why not present it as such? Why is the probe delay limited to 4 ps, and why waste half of each PPP frame on negative times which teach us nothing? Seeing that the cooling time for InP seems slowest, wouldn't the authors do well to extend the delay to verify cooling is complete? Is it a coincidence that the remainder of non-restored bleach is only observed in the most noisy PPP data, which probably indicates very weak push effects?

The PPP data is presented as a single channel signal, but the experiment should provide a broad band difference spectrum. Have the authors followed the push induced Δa during the cooling? Does it resemble the spectral change observed for cooling following simple PP exciting in the blue? Is the PPP signal the result of integrating over a range of wavelengths, or truly a single channel result? Not learning from the PPP spectrum seems strange.

For all the above reasons I find this paper not suitable for publication at this time.

Version 3:

Reviewer comments:

Reviewer #4

(Remarks to the Author)

It is obvious that the authors have seriously considered the criticism of all reviewers, including my own, and as a result the paper is more balanced. None the less I am not convinced by the authors arguments for the following reasons. I find the perfect match between the levels of signal saturation, full transparency before Auger recombination, and a reduction to 50% bleach - exactly as reported decades ago for CdSe cores, just too much of a coincidence to arise from a totally different mechanism here. I note that work of colleagues who are authors here has also shown time and again that carrier cooling in virtually all semiconductor quantum dots leads not only a buildup of the lowest exciton bleach, but also to significant changes in the pump-probe spectrum, including disappearance of state filling bleaches higher up, and appearances of both net absorption as well as bleaches, and in particular an induced absorption below the band gap. I therefore wonder how they accept that the push is exciting the electrons to higher levels but for some reason causes no similar effects on the bleach spectrum during the cooling which follows. I would therefore recommend that the authors at least comment on these unusual points, but leave that option to their discretion. Thus I find the paper acceptable for publication as is, with the recommendation above. I rest my case.

Response to the Nature Communications Reviewer Reports for Vonk *et al.* (NCOMMS-24-20930)

We thank the Reviewers for their detailed reading of our work and their constructive comments, which were very helpful for further improvement of the manuscript. We have made revisions in response to their reports, as detailed below. Most importantly, in the revised version of the manuscript, we have added new experiments and analysis. These lead to a hypothesis on the nature of the hot-carrier trapping site and to possible synthesis strategies to mitigate hot-carrier trapping. The Reviewers' comments are reproduced in blue, followed by our response. Our modifications of and additions to the manuscript text are reproduced in red.

Reviewer 1:

This study delves into the limitations of InP QDs, particularly focusing on hot carrier trapping and delayed fluorescence dynamics. While the investigation of hot carrier trapping via pump-probe-push spectroscopy is intriguing, the correlation drawn between delayed fluorescence and trapped hot carriers lacks substantial evidence.

Our response: We gratefully thank Reviewer 1 for his/her interest in our finding of hot-carrier trapping in InP-based QDs.

It is unfortunate that we were unable to convey the experimental proof we found for the causal link between hot-carrier trapping and delayed fluorescence. Indeed, the evidence from the ensemble measurements presented in Figs. 2,3 is only circumstantial. This is why we performed single-particle measurements in Fig. 4. These provide direct proof for a cause–effect relationship between hot-carrier trapping and delayed fluorescence.

In the single-particle experiments, we synchronously follow the excited-state dynamics and emission spectrum of a single InP/ZnSe/ZnS QD in time. As is typical for single QDs, their properties fluctuate in time. We can thus compare periods of significant hot-carrier trapping with periods of negligible hot-carrier trapping on the same QD. The QD of Fig. 4 shows hot-carrier trapping between $T = 2\text{--}4$ s in Fig. 4c,d. During such moments the short lifetime component has a lower amplitude, the hallmark proof of hot-carrier trapping. Simultaneously, a longer lifetime component with redshifted spectrum appears, evidencing slow trap emission. Compare Fig. 4f (selected periods with no hot-carrier trapping) to Figs. 4g,h.

Action 1: In the revised manuscript, we write more clearly that Figs. 2,3 provide only circumstantial evidence for the relation between hot-carrier trapping and delayed emission. We have removed panel b from Fig. 3 that shows inconclusive results on the spectral characteristics of delayed emission. In the revised manuscript, we more clearly write that definite proof for the cause–effect relationship and recombination mechanism follow from single-particle experiments in Fig. 4:

Introduction:

“In this Article, we combine ensemble and single-particle experiments to investigate why InP-based QDs are not yet suitable for lasing applications, despite their high brightness, and how their performance may be improved. On the ensemble level, we find a correlation between the magnitude of charge-carrier losses on the sub-ps timescales and slow delayed emission on the ns-to- μ s timescales. From single-particle measurements, we find a cause–effect relationship between hot-carrier trapping and delayed trap emission. “

We rephrased the discussion surrounding Fig. 3, to convey that the match alone of trapping probability with delayed emission probability is no hard evidence for a causal link, but the single-particle measurements are essential:

“The match between the hot-carrier-trapping probability (Fig. 2a, $P_{\text{trap}} = 11.8 \pm 0.7\%$) and fraction of delayed emission (Fig. 3a, $13.8 \pm 0.7\%$) suggests a cause-and-effect relation, although this match alone is not enough to draw a solid conclusion.”

and

“In the next section, single-particle measurements will turn out ideal to establish the cause-and-effect relation between hot-carrier trapping and delayed emission.”

We updated the headings of the sections discussing Fig. 3 and Fig. 4 and the Figure titles of Figs. 3,4; to emphasize which experimental results show correlations and which causation:

Section header and title of Fig. 3 changed to: “**Correlation between hot-carrier trapping and delayed emission**”

Section header and title of Fig. 4 changed to : “**Cause–effect relation between hot-carrier trapping and delayed trap emission**”

We updated Fig. 3 without delayed-emission spectrum from ensemble measurement:

The authors attribute three time constants extracted from transient absorption results to the decay of various excitonic states. However, this assertion lacks robust support and contradicts previous discussions regarding the timing of the Auger phase.

Our response: We agree with the Reviewer that the assignment of the fast decay components to biexcitons and triexcitons was not thoroughly supported. In the revised manuscript, we included new analysis of the power-dependent transient-absorption measurements to justify our assignment.

(Multi)exciton formation follows Poissonian absorption statistics. As such, the formation probability of biexcitons should follow $p_{\text{BX}} \propto (\bar{n})^2$ in the limit of small excitation number \bar{n} . We investigate the formation of biexcitons and higher multiexcitons in our pump–probe measurements by fitting the power-dependent band-edge bleach traces ($\bar{n} = 0-7$) to a model of triexponential decay. More specifically, we perform a global-fit procedure for slow τ_{long} , intermediate τ_{int} , and short τ_{short} lifetime components, with different amplitude contributions to each transient-bleach trace. Panel **a** below shows that such a model fits the data nicely for $\tau_{\text{int}} = 147$ ps and $\tau_{\text{short}} = 29$ ps, while the fitted τ_{long} is much longer than the time range of the measurement (1000 ps) and cannot be determined accurately. Panel **b** below shows the amplitudes of the long A_{long} (blue), intermediate A_{int} (green), and short A_{short} (red) lifetime components as a function of excitation number \bar{n} on a log–log scale. We observe that A_{long} increases linearly (blue solid line) and A_{int} increases quadratically (green solid line) with \bar{n} , consistent with the formation of excitons and biexcitons, respectively. Moreover, the short-lifetime component A_{short} follows an even higher-order dependence on \bar{n} (red solid line for cubic dependence), consistent with the formation of higher multiexcitons. These results confirm our previous attribution of the fast-decaying lifetime components to the decay of biexcitons ($\tau_{\text{int}} = \tau_{\text{BX}} = 147$ ps) and multiexcitons ($\tau_{\text{short}} = \tau_{\text{MX}} = 29$ ps). The biexciton lifetime is on the same order of magnitude but $2\times$ longer than reported in previous studies on InP/ZnSe QDs, highlighting that our sample has somewhat suppressed Auger recombination compared to previous studies ([J. Phys. Chem. C 2021, 125, 15405–14414] and [Adv. Opt. Mater. 2022, 10, 202200328]).

Action 2: We have included new analysis of the transient-absorption dynamics as a function of excitation power. The power-dependent fitting results are shown in the Supplementary Information.

Supplementary Fig. S1 | Power-dependent lifetime analysis. (a) Transient band-edge bleach traces as a function of excitation number $\bar{n} = 0-7$ (blue to red). Black lines: global fit of triexponential decay to all traces including long τ_{long} , intermediate τ_{int} , and short τ_{short} lifetime components, with varying amplitude contributions. We find $\tau_{\text{int}} = 147$ ps and $\tau_{\text{short}} = 29$ ps. The fitted τ_{long} is unphysically long, which we attribute to exciton decay (several tens of ns) much slower than the timescales of the pump–probe experiment (<1 ns). (b) Fitted amplitudes of the long (blue dots), intermediate (green dots), and long (red dots) lifetime components as a function of excitation number \bar{n} . The long and intermediate components follow a linear (blue line) and quadratic (green line) increase with \bar{n} , consistent with Poissonian absorption statistics of excitons and biexcitons, respectively. The short-lifetime component shows an even higher-order increase with \bar{n} (red line shows cubic increase), indicating the formation of higher multiexcitons. Based on this we attribute the fast-decaying lifetime components to biexcitons ($\tau_{\text{int}} = \tau_{\text{BX}} = 147$ ps) and higher multiexcitons ($\tau_{\text{short}} = \tau_{\text{MX}} = 29$ ps). The biexciton lifetime is $2\times$ longer than reported in previous studies on InP/ZnSe/ZnS QDs^{S1,S2}.

(S1) Nguyen, A.T.; La Plante, I.J.; Ippen, C.; Ma, R.; Kelley, D.F. Extremely Slow Trap-Mediated Hole Relaxation in Room-Temperature InP/ZnSe/ZnS Quantum Dots *J. Phys. Chem. C* 2021, 125, 4110–4118.

(S2) Sousa Velosa, F.; Van Avermeat, H.; Schiettecatte, P.; Mingabudinova, L.; Geiregat, P.; Hens, Z. State filling and stimulated emission by colloidal InP/ZnSe core/shell quantum dots *Adv. Opt. Mater.* 2022, 10, 2200328.

We refer to this additional Figure in the Supplementary Information in the main text:

“We attribute the fitted time constants of 147 ps and 29 ps to the decay of biexcitons and higher multiexcitons, respectively (Supplementary Information Fig. S1)^{21,23}.”

(21) Sousa Velosa, F. *et al.* State filling and stimulated emission by colloidal InP/ZnSe core/shell quantum dots. *Adv. Opt. Mater.* **10**, 202200328 (2022).

(23) Nguyen, A. T., Plante, I. J. La, Ippen, C., Ma, R. & Kelley, D. F. Extremely Slow Trap-Mediated Hole Relaxation in Room-Temperature InP/ZnSe/ZnS Quantum Dots. *J. Phys. Chem. C* **125**, 4110–4118 (2021).

The discussion on the dynamic behavior of the SE peak in the transient absorption spectrum would benefit from the inclusion of the actual spectrum itself. Without visual aid, the clarity of the analysis is compromised, leaving readers to speculate on the nature of the observed dynamics.

Action 3: We have included the time-resolved gain spectrum at the highest excitation power ($\bar{n} = 7.1$) to Fig. 5 of the main text to clearly visualize the time and color dynamics of the gain. Additionally, we add time-resolved gain spectra at different \bar{n} to the Supplementary Information.

“(b) Time-resolved gain spectra (masked excited-state absorption $A < 0$) for $\bar{n} = 7.1$. We observe Stokes-shifted gain up to 200 ps after photoexcitation. The photon energy of maximum gain slightly redshifts with time as \bar{n} decreases through electron–hole recombination, as expected from our model that considers the Stokes shift between stimulated emission and absorption introduced in a.”

“Our adapted hot-carrier trapping model reproduces the gain spectrum directly after cooling (Figs. 5a,c) and the maximum optical gain (Fig. 5d) as a function of \bar{n} . In line with these observations, time-resolved gain spectra reveal that the gain redshifts over the course of the recombination Auger phase (0–200 ps) as the charge carrier population decreases (Fig. 5b and Supplementary Information Fig. S13).”

Supplementary Fig. S13 | Time-dependent gain spectra of InP/ZnSe/ZnS QDs. (a) Time-dependent gain (masked excited-state absorption $A < 0$) spectra at $\bar{n} = 1.4$. (b)–(d) Same as a, but for $\bar{n} = 2.7$, $\bar{n} = 4.5$, and $\bar{n} = 7.1$. We observe that the gain blueshifts and increases in magnitude as a function of \bar{n} , but for each experiment redshifts as a function of time after excitation.

Fig. 2 b-e presents data on the photoluminescence quantum yields (PLQYs) of reference QDs with different compositions. However, the text fails to provide these values, leaving readers without crucial information necessary for understanding the significance of the presented results.

Our response: We agree with the Reviewer that it is important to give numbers for the performance of the different QD compositions in our work. We should also more clearly explain the relation of the experiments in Fig. 2b–e to the PLQY of our samples.

Fig. 2b–e shows the hot-carrier trapping process. For the high-quality InP/ZnSe/ZnS sample studied in this work, the hot-carrier trapping probability (12%) is higher than the overall nonradiative losses (5% = $1 - \text{PLQY}$). This evidences that hot carrier trapping does not necessarily lead to loss of emission. Indeed, the further figures (Fig. 3,4) reveal delayed emission due to trapped charges. Hot carrier trapping delays the emission but does not necessarily quench it.

Action 4: In the discussion surrounding Fig. 2, we now clarify the storyline by highlighting the seeming inconsistency of a higher hot-carrier trapping probability (12%) than fluorescence quenching (5%):

“Importantly, the hot-carrier trapping probability of $P_{\text{trap}} = 11.8 \pm 0.7\%$ is higher than the fluorescence quenching probability under low-power illumination, which amounts to $1 - \text{PLQY} = 5\%$. This evidences that hot-carrier trapping does not necessarily lead to fluorescence quenching. Below we will investigate why.”

We now provide the PLQY values of all materials and extracted trapping probabilities P_{trap} (including fit uncertainty) of the InP-based QDs to the caption of Fig. 2. The PLQY values of some samples are estimated literature values. We provide the trapping probabilities of the other materials to Supplementary Note 2.

"Fig. 2 | Identifying hot-carrier losses in different QD materials. (a) Bleach transient of the band-edge absorbance of our InP/ZnSe/ZnS QDs (PLQY = 95%, $P_{\text{trap}} = 11.8 \pm 0.7\%$) in a pump–push–probe experiment. (I) A 3.1-eV pump laser pulse at $t = -300$ ps generates $\bar{n} = 0.1$ excitations. (II) A 0.95-eV push laser pulse (push fluence of 4.6 mJ cm^{-2}) excites a fraction of the band-edge electrons (Supplementary Fig. S4) to a hot-carrier state, decreasing the absorbance bleach. (III) 88% of the hot charge carriers cool down to the band edge, but 12% of the carriers are lost due to hot-carrier trapping. Solid blue line: fit to an exponential cooling model convolved with the instrument-response function, yielding a time constant of 795 fs. Grey line: deconvolved PPP trace. The y -axis is normalised, setting the bleach just before the push to 1 and immediately after the push to 0. **(b–e)** Similar PPP experiments on **b** CdSe/CdS/ZnS QDs (PLQY = 95%), **(c)** CsPbBr₃ nanocrystals (estimated PLQY = 50–90%⁷), **(d)** CuInS₂/CdS QDs (estimated PLQY = 90%³⁶), and **(e)** “bare” InP/ZnSe (estimated PLQY = 50%³⁵, $P_{\text{trap}} = 23.6 \pm 0.9\%$), without a ZnS shell. Only the InP-based samples (**a** and **e**) show significant hot-carrier losses, while in the CdSe, CuInS₂, and perovskite-based materials, the bleach recovers by 97% or better (Supplementary Note 2 for precise values for P_{trap}).”

(7) Protesescu, L. et al. Nanocrystals of cesium lead halide perovskites (CsPbX₃, X = Cl, Br, and I): novel optoelectronic materials showing bright emission with wide color gamut. *Nano Lett.* **15**, 3692–3696 (2015).

(35) Tessier, M.D. et al. Economic and Size-Tunable Synthesis of InP/ZnE (E = S, Se) Colloidal Quantum Dots. *Chem. Mater.* **27**, 4893–4898 (2015).

(36) Berends, A.C. et al. Radiative and Nonradiative Recombination in CuInS₂ Nanocrystals and CuInS₂-Based Core/Shell Nanocrystals. *J. Phys. Chem. Lett.* **23**, 3503–3509 (2016).

Supplementary Note 2:

“Using this fitting procedure, we find hot-carrier probabilities for CdSe/CdS/ZnS (Fig. 2b, $P_{\text{trap}} = 3.4 \pm 1.6\%$), CsPbBr₃ (Fig. 2c, $P_{\text{trap}} = 1.8 \pm 1.5\%$), and CuInS₂/CdS (Fig. 2d, $P_{\text{trap}} = 0.8 \pm 0.3\%$). These small but nonzero trapping probabilities might arise due to minute power instability of the pump laser over the PPP experiment, resulting in higher (or lower) bleach as different pump–probe delays are scanned. This experimental factor complicates quantifying trapping probabilities close or equal to zero. “

The authors posit that despite the high probability of hot carrier trapping in InP QDs, the achievement of high PLQY is attributed to "delayed fluorescence." However, the assertion that the probability of hot carrier trapping is comparable to the amplitude of delayed fluorescence lacks substantiation.

See **Action 1**.

Furthermore, there is a conspicuous absence of explanation regarding the distinction between "delayed fluorescence" and conventional trap emission, as well as how this phenomenon contributes to the enhancement of PLQY in the QDs. Clarification on these points is essential for a comprehensive understanding of the proposed mechanism.

Our response: We thank the Reviewer for pointing this out. Indeed, “delayed” emission (or delayed fluorescence) has become a collective term for all possible photophysical mechanisms that lead to slow emission compared to conventional band-edge emission. Slow emission can be due to direct recombination of a trapped charge carrier, due to detrapping followed by radiative recombination of delocalized carriers, or due to polarization of the exciton by fluctuating external electric fields (quantum-confined Stark effect).

In this work, we do not wish to convey that the involvement of hot-carrier traps would somehow **enhance** the PLQY. Instead, we find that the PLQY is **preserved despite** hot-carrier traps. This challenges the common picture that trapping leads to a decrease of the photoluminescence quantum yield (PLQY), such as found for CdSe-based QDs due to band-edge trapping [ACS Nano 2018, 12, 3397–3405] or hot-carrier trapping [Nature 2011, 479, 203–207]. The high PLQY of InP/ZnSe/ZnS QDs is preserved, although the gain is negatively affected, because the emission occurs on a longer (= delayed) timescale.

Based on this comment and a comment by **Reviewer 2**, we were motivated to delve deeper into the microscopic origin of the trap state. In our single-particle experiments, we find that the spectrum of trap emission is different from band-edge emission. However, the spectral shift and difference in linewidth are significantly smaller than for typical surface-related trap emission, such as from core-only CdSe QDs [Phys. Rev. B 2013, 87, 081201]. Moreover, the excited-state lifetime of the trap emission in InP/ZnSe/ZnS QDs is longer than the band-edge emission, but not by the factor of ~10 observed for CdSe nanoplatelets [ACS Nano 2021, 15, 7216–7225]. Based on these observations, we hypothesize that the hot-carrier traps in InP/ZnSe/ZnS QDs are not located on the outer surface but on the InP/ZnSe core–shell interface.

The redshift and broadening of trap emission compared to band-edge emission are due to vibrational coupling. Weaker vibrational coupling is expected for internal trapping than for surface trapping. Indeed, the interior of a QD is simply more rigid than the outer surface [Nature 531, 618–622 (2016)]. Moreover, Fröhlich interaction with polar phonons depends on charge separation, which is smaller when a charge is trapped internally compared to the QD surface [Nano Lett. 16, 289–296 (2016)]. Finally, the radiative recombination rate scales with electron–hole overlap, which is larger following internal trapping compared to surface trapping.

The (1) minor redshift, (2) minor broadening, and (3) minor slow-down of our trap-state emission compared to band-edge emission is therefore consistent with trapping at an internal defect, for example at the InP/ZnSe core–shell interface.

To test this hypothesis and to further characterize the trap state, we performed transient-PL measurements comparing excitation into the ZnSe shell ($E_{\text{laser}} = 3.1$ eV, panel **b** below) and the InP core ($E_{\text{laser}} = 2.4$ eV, panel **c** below). We observe that core (12%) and shell excitation (14%) yield similar contributions of delayed emission to the total PL. If the trap state were located on the QD surface, we would expect a significantly larger delayed-emission contribution for shell excitation as the photogenerated hot charge carriers could reach the surface more easily. This further confirms the idea of internal defect sites responsible for hot-carrier trapping.

The hypothesis is also consistent with the difference between the two InP-based QD samples investigated in this work. The high-quality sample that is the main focus of this work, underwent an interface treatment during the synthesis that oxidises the surface of the InP core before shell growth [ACS Nano 16, 9701–9722 (2022)]. The trapping probability and delayed emission contribution are 2× lower than the sample synthesised following an older procedure without interface treatment [Chem. Mater. 27, 4893–4898 (2015)].

Despite these hints and experimental support for the nature of the trap state, we realise that our identification of the trap locations is tentative. This is why we phrase our discussion in the main text carefully, using terms such as “propose” and “indicative”.

Action 5: We now discuss the hypothesis of internal defects on the core–shell interface leading to hot-carrier trapping and delayed emission to the manuscript.

Introduction:

“Based on the characteristics of the trap-related emission, we propose that hot-carrier traps are most likely internal defects, for example located on the InP/ZnSe interface. This highlights the direction into which InP-based QDs should be improved for next-generation applications.”

Discussion section of Fig 4:

“The characteristics of the trap emission are most consistent with hot-carrier trapping due to internal defects, for example at the InP/ZnSe core–shell interface. Trap emission from surface traps is typically strongly broadened (linewidths of 0.3 eV) and redshifted (by up to 0.6 eV) compared to band-edge emission⁴⁸. In contrast, the trap emission from our InP-based QDs has a relatively narrow linewidth and minor redshift compared to the band edge (Fig. 4)⁴⁸. The redshift and broadening of trap emission are due to vibrational coupling, which is expected to be weaker for an internal trap compared to a surface trap (Supplementary Fig. S16). Indeed, the interior of a QD is simply more rigid than the outer surface⁴⁹. Moreover, Fröhlich interaction with polar phonons depends on charge separation, which is smaller when a charge is trapped internally compared to the QD surface⁵⁰. The minor broadening and redshift of our trap emission is thus indicative of internal trapping. Comparison of the trap emission intensity (Supplementary Fig. S15) for different excitation photon energies confirms this: the trapping probability hardly depends on the excitation energy, while a surface trap would be more accessible upon excitation at energies beyond the shell bandgap.”

(48) Mooney, J., Krause, M. M., Saari, J. I. & Kambhampati, P. Challenge to the deep-trap model of the surface in semiconductor nanocrystals. *Phys. Rev. B - Condens. Matter Mater. Phys.* **87**, 1–5 (2013).

(49) Bozyigit, D. *et al.* Soft surfaces of nanomaterials enable strong phonon interactions. *Nature* **531**, 618–622 (2016).

(50) Cui, J. *et al.* Evolution of the Single-Nanocrystal Photoluminescence Linewidth with Size and Shell: Implications for Exciton-Phonon Coupling and the Optimization of Spectral Linewidths. *Nano Lett.* **16**, 289–296 (2016).

Action 6: We present additional trap-state characterization to the Supplementary Information:

Supplementary Fig. S15 | Dependence of delayed emission on InP core vs. ZnSe shell excitation. (a) Absorbance spectrum of the InP/ZnSe/ZnS QD sample. For photon energies $E > 2.84$ eV, the absorbance increases because of additional absorption by the ZnSe shell material. Below that photon energy, we mainly have absorption by the InP core. (b) Transient-PL measurement of the InP/ZnSe/ZnS QDs upon excitation into the ZnSe shell, at an energy of $E_{\text{laser}} = 3.1$ eV. (c) Same as b, but for 515-nm excitation ($E_{\text{laser}} = 2.4$ eV).

Action 7: We now present an overview of delayed-emission characteristics due to different emission mechanisms in the Supplementary Information:

Supplementary Fig. S16 | Possible mechanisms of trap-related delayed emission. (a) A delocalised excitation in InP-based QDs gives rise to band-edge emission (middle) and decay (bottom). (b) Charge-carrier detrapping following by recombination of delocalised carriers yields the same emission spectrum as a. The decay dynamics (red) change as the emitting population is first fed by detrapping after which electron–hole recombination leads to PL. (c) Trap emission from a delocalized hole and an electron trapped on the QD surface. Because of the large reorganization energy and large static dipole moment, the emission spectrum is significantly redshifted and broadened compared to regular emission from a delocalized excitation. (d) Trap emission from a delocalized hole and a trapped electron on the core–shell interface. Here, we expect smaller reorganization energies and less phonon broadening. This combination of signatures is consistent with all our experimental results.

Action 8: In light of the new discussion on the trap-state location—based on additional characterisation of the trap state—we revise the discussion on the difference between P_{trap} for InP/ZnSe/ZnS QDs and InP/ZnSe QDs.

“In contrast, the hot-carrier losses are even higher (Fig. 2e, $P_{\text{trap}} = 23.6 \pm 0.9\%$) for InP/ZnSe QDs synthesized following the procedure introduced by Tessier *et al.*³⁵. This might highlight the importance of the additional interface oxidation treatment used for the InP/ZnSe/ZnS QDs used in this work¹⁰, which possibly reduces defects responsible for hot-carrier trapping on the InP/ZnSe interface.”

(10) Van Avermaet, H. *et al.* Full-spectrum InP-based quantum dots with near-unity photoluminescence quantum efficiency. *ACS Nano* **16**, 9701–9712 (2022).

(35) Tessier, M. D., Dupont, D., De Nolf, K., De Roo, J. & Hens, Z. Economic and Size-Tunable Synthesis of InP/ZnE (E = S, Se) Colloidal Quantum Dots. *Chem. Mater.* **27**, 4893–4898 (2015).

Reviewer 2:

The manuscript entitled "Hot-Carrier Trapping Preserves High Quantum Yields but Limits Optical Gain in InP-Based Quantum Dots" by Freddy Rabouw *et al.* investigates the photophysics of Indium Phosphide (InP)-based quantum dots (QDs), focusing on the phenomenon of hot-carrier trapping. They claim that the trapping maintains high quantum yields but limits optical gain, making these QDs unsuitable for high-intensity applications like lasers. The paper presents comprehensive evidence for the idea of hot-carrier trapping, combining both ensemble-based and single-quantum-dot spectroscopy to understand the limitations of InP-based QDs. They also used advanced spectroscopic techniques, such as time-resolved spectroscopy and pump-push-probe experiments, to provide robust and detailed insights into the hot-carrier dynamics. The comparison with other quantum dot materials effectively highlights the unique challenges associated with InP-based QDs.

Our response: We thank the Reviewer for his/her positive view on our experimental results.

However, some experimental results need to be further validated, especially the pump-push-probe part, which plays a key role in the whole paper. For these reasons, I recommend a major revision before considering this work for publication.

My concerns are as follows:

One of my concerns is about the pump-push-probe results, which provide direct evidence of hot-carrier trapping in the manuscript. The pump-push-probe is a complicated experimental technique that involves three-pulse interaction. In this paper, a near-infrared pulse (0.95 eV, 1.3 μm) was used as a push to excite the cold carrier to the hot carrier population. The cold carrier absorption of photons requires assistance from phonons, making the absorption cross-section very low. For this reason, the push pulse always requires high fluence to observe hot-carrier excitation. Under high push fluence, the quantum dot system is very likely to have interband gap transitions due to multiphoton absorption. When there is interband transition induced by the push pulse, it will result in a reduced bleach signal, which will be similar to the observed “hot-carrier trap.” The authors also reported different push pulse energies (0.51 eV) observing different results, which could also be due to the lower multiphoton absorption cross-section at lower energy. To exclude this possibility, I recommend providing more details of experiments such as push fluence, and conducting multiphoton absorption experiments for all the samples to confirm the multiphoton absorption threshold.

Our response: We fully understand the reviewer’s concern. Indeed, one needs to be careful and distinguish the desired push effect (*i.e.* optical reheating of cold carriers) from interband multiphoton absorption. Below we describe our experimental design in more detail and perform additional data analysis to rule out interfering contribution from multiphoton effects.

In our pump–push–probe experiment design, an optical chopper wheel modulates the pump beam at 500 Hz, while the push and probe beams are operated at the 1 kHz repetition rate of the pump laser. Therefore, the differential absorbance $A_{\text{pump on, push on, probe on}} - A_{\text{pump off, push on, probe on}}$, *i.e.* the pump–push–probe signal, should not directly contain push–probe contributions such as multiphoton absorption.

Furthermore, we note that the first-order effect of multiphoton absorption would be an *increased* bleach due to the push pulse, because of additional band-edge charge carriers. In contrast, the pump–push–probe experiments show a *reduced* bleach due to the push pulse, which evidences charge-carrier losses. The difference can be observed in the revised Supplementary Figure S6. Hence, while multiphoton absorption should be carefully considered, it would not be a good alternative explanation of (part of) the observed signal.

Nevertheless, we characterised the multiphoton absorption of our semiconductor QDs in more detail. We performed pump–push–probe experiments at increasing push fluence and with different push energies. The results are shown below (Supplementary Fig. S6). The apparent hot-carrier losses are constant at low fluence below 10 mJ cm^{-2} but increase for higher push fluences. The increased hot-carrier losses for higher fluences must be an indirect effect of interband multiphoton absorption, because the direct effect of interband absorption is the generation (rather than loss) of carriers. Indeed, control experiments with the pump pulse blocked show significant three-photon-induced bleach at push powers exceeding 10 mJ cm^{-2} .

In our experiments in the main text, we use a push fluence of 4.6 mJ cm^{-2} . This is well below the regime where multiphoton absorption affects charge-carrier losses ($>10 \text{ mJ cm}^{-2}$). The effect of any multiphoton absorption is subtracted in our experimental procedure, as explained above. As a visual demonstration of interband multiphoton absorption, the revised Fig. S6 in the Supplementary Information (containing some data from Fig. S5 of the first submission) compares the observed double-differential absorbance in the pump–push–probe experiment with the bleach signal in a control experiment with the pump pulse blocked, for two push photon energies. We also added a new Supplementary Fig. S7 of similar check experiments on the other semiconductor materials depicted in Fig. 2b–d of the main text.

Action 9: Supplementary Fig. S6 is updated with control experiments of interband multiphoton absorption. Some of the plots were previously part of Supplementary Fig. S5, while panels **d–f** are entirely new.

Supplementary Fig. S6 | Hot-carrier losses and multiphoton absorption in pump-push-probe experiments. (a) Pump-push-probe (PPP) experiments of the InP/ZnSe/ZnS QD sample using push pulse energies of 0.95 eV (blue dots) and 0.52 eV (red dots). We observe larger hot-carrier losses for the high-energy push pulse (12%) compared to the low energy push (9%). (b) Hot-carrier losses as a function of push fluence for both 0.95 eV (blue) and 0.52 eV (red) push energies. For small push fluences, the hot-carrier losses are independent of push power for both push energies. At higher push powers, the extracted hot-carrier losses increase. (c) Absorbance bleach in push-probe experiments with a sub-bandgap push pulse ($E_{\text{push}} = 0.95$ eV) and no prior pump pulse on InP/ZnSe/ZnS QDs for varying push fluences. Dashed blue line: absorbance bleach due to the 3.1-eV pump pulse at the fluence used in the PPP experiments of the main text. The absorbance bleach shows a cubic dependence on push fluence (blue solid line) showing that 3-photon absorption populates the band-edge. In the PPP experiments, hot-carrier losses increase with push power, which might be due to the creation of multiexcitons by multi-photon absorption followed by positive feedback on losses, as discussed in Supplementary Note 1. (d) Transient double-differential absorbance for a PPP experiment (blue, 2.6 mJ cm^{-2} push fluence), which is the difference between the PPP absorbance bleach and the pump-probe (PP) absorbance bleach. We investigate the effect of the push without pump by measuring the push-probe trace at 7 mJ cm^{-2} (pump blocked). We reconstruct the push-probe trace at 2.6 mJ cm^{-2} (grey) using the cubic fluence dependence of 3-photon absorption to scale the data. The derivative-like feature at the arrival of the push probe is due to the optical Stark effect. For push-probe delays $t > 5$ ps we observe a small negative double-differential absorbance, which indicates multiphoton absorption induced by the push pulse. In a PPP experiment, we measure the differential absorbance, *i.e.* $A_{\text{pump on, push on, probe on}} - A_{\text{pump off, push on, probe on}}$, so the PPP signal does not directly contain push-probe effects such as multiphoton absorption or the optical Stark effect. (e) Same as **d**, but for a push fluence of 4.6 mJ cm^{-2} . The push-probe transient at 4.6 mJ cm^{-2} (grey) is again reconstructed from a 7 mJ cm^{-2} push-probe experiment. We observe more interband multiphoton ($t > 5$ ps) absorption induced by the push pulse. However, we observe almost equal hot-carrier losses as for the lowest push fluence (see panel **b,d**), showing the robustness of the differential-absorbance experiment design. (f) Same as **d,e**, but for a 0.52-eV push energy and a push fluence of 10.2 mJ cm^{-2} (red). Reconstructed push-probe signal at 10.2 mJ cm^{-2} (grey), reconstructed from a measurement at a push fluence of 12 mJ cm^{-2} . Here, we use that a band-edge excitation is formed by 5-photon absorption. We observe no multiphoton absorption in this measurement.

Action 10: We include double-differential traces and push-probe traces of the other QD samples to the Supplementary Information (Fig. S7) to highlight the small multiphoton absorption by the push pulse:

Supplementary Fig. S7 | Hot-carrier losses and multiphoton absorption of other semiconductor materials. (a) Transient double-differential absorbance of CsPbBr₃ nanocrystals (blue, 5.0 mJ cm⁻² push fluence) showing negligible hot-carrier losses. Grey: push-probe trace (push fluence 5.0 mJ cm⁻²) showing minor multiphoton absorption induced by the push. (b)–(d) Same as a, but for CdSe/CdS/ZnS b, CuInS₂/CdS c, InP/ZnSe d. In all cases, the bleach due to multiphoton absorption is small compared to the bleach recovery in PPP experiments.

The paper describes delayed emission attributed to hot-carrier trapping. Can the authors provide more insights into the nature of the trap states?

See **Action 5**.

The authors claim that hot-carrier trapping limits optical gain. Will resonant excitation induce lasing in this material?

Our response: This is an interesting suggestion, which we should address in the manuscript. Indeed, resonant excitation in principle circumvents hot-carrier trapping as the band-edge states are directly populated by optical excitation. However, resonant optical excitation cannot pump a system any further than to optical transparency. At increasing excitation fluence, the excitation pulse itself would induce stimulated emission, limiting the achievable gain to zero. Exactly this is why QDs, with their multi-level energy structure, are so interesting for lasing applications.

Action 10: We added the potential solution of resonant excitation to circumvent hot-carrier trapping to the discussion section of the main text:

“A clever solution to circumvent hot-carrier trapping may be resonant excitation of the QDs. However, one would then operate the QD as a 2-level system and, unfortunately, the fundamental laws of stimulated emission make population inversion by optical excitation of a 2-level system impossible⁵¹. The possibility of nonresonant optical excitation is exactly what makes QDs so interesting for lasing.”

(51) Svelto, O. *Principles of Lasers*. (Springer, 2010).

Additionally, the paper does not sufficiently explore potential methods to mitigate hot-carrier trapping in InP-based QDs, which limits the practical applicability of the findings. Can the authors propose specific synthesis modifications or experimental techniques that might mitigate hot-carrier trapping?

Our response: Following the suggestions of **Reviewer 1 and 2** we discovered more about the microscopic origin of the trap state (**Action 5**). Based on this, we can propose strategies to mitigate hot-carrier trapping. We added this discussion to the main text. As the trap state is likely on the InP/ZnSe core-shell interface, a potential avenue to circumvent trapping might be to polarise the InP/ZnSe interface by synthesizing In-terminated InP cores [Nat. Mater. 21, 246–252 (2021)]. Here, outward-pointing bond dipoles between In and Se increase the potential-energy depth for electrons

in the InP core, decreasing the electron–trap state separation. Other strategies might circumvent formation of the trap state all together, for example by using shell materials with better matching lattice constants to InP, such as MgSe [ACS Appl. Nano Mater. 3, 3859–3867 (2020)]. The difference in hot-carrier trapping probability between the sample of Fig. 2a, 3a and the sample of Fig. 2e, 3d shows a beneficial effect of an oxidation treatment of the core surface during the synthesis.

Action 11: We added potential strategies to prevent hot-carrier trapping in InP-based QDs to the Discussion section of the revised manuscript:

“As the trap emission observed is most consistent with internal traps, the internal structure of the QDs seems the most important target. A potential avenue lies in wave-function engineering by interface polarization⁵³. InP cores can form polarized bonds with the ZnSe shell material that yield inward or outward-pointing dipole moments depending on the InP surface termination. Outward dipole moments could distance the electron wave function from potential interfacial traps on the core–shell interface and prevent hot-carrier trapping. Other strategies could attempt to circumvent the formation of internal and interfacial trap states all together, for example by using MgSe⁵⁴ shelling, which lowers the lattice mismatch between the core and shell material. The difference in hot-carrier trapping probability between the sample of Fig. 2a, 3a and the sample of Fig. 2e, 3d shows a beneficial effect of an oxidation treatment of the core surface during the synthesis^{10,35}.”

(10) Van Avermaet, H. *et al.* Full-spectrum InP-based quantum dots with near-unity photoluminescence quantum efficiency. *ACS Nano* **16**, 9701–9712 (2022).

(35) Tessier, M. D., Dupont, D., De Nolf, K., De Roo, J. & Hens, Z. Economic and Size-Tunable Synthesis of InP/ZnE (E = S, Se) Colloidal Quantum Dots. *Chem. Mater.* **27**, 4893–4898 (2015).

(53) Jeong, B. *et al.* Interface polarization in heterovalent core–shell nanocrystals. *Nat. Mater.* **21**, 246–252 (2021).

(54) Mulder, J. T. *et al.* Developing Lattice Matched ZnMgSe Shells on InZnP Quantum Dots for Phosphor Applications. *ACS Appl. Nano Mater.* **3**, 3859–3867 (2020).

Some experimental details, such as the exact conditions of the spectroscopic measurements (e.g., transient absorption spectrum, pump-push-probe excitation fluence) and fitting parameter uncertainty, are not fully elaborated, which might hinder reproducibility.

See **Action 4** for added fit uncertainties in P_{trap} .

Action 12:

We added the push fluence to the caption of Fig. 2:

“(II) A 0.95-eV push laser pulse (push fluence of 4.6 mJ cm^{-2}) excites a fraction of the band-edge electrons (Supplementary Fig. S3) to a hot-carrier state, decreasing the absorbance bleach.”

We add the fit uncertainty of the delayed-emission contribution P to the caption of Fig. 3:

“We find a relative contribution of delayed emission of $P = 13.8 \pm 0.7\%$, close to the hot-carrier losses found in Figs. 1,2. ($P_{\text{trap}} = 11.8 \pm 0.7\%$).”

“(d) Same as a, but for bare InP/ZnSe QDs without a ZnS outer shell, which clearly shows more delayed emission ($P = 35.5 \pm 1.5\%$).”

Overall, while the study provides significant insights into the limitations of InP-based QDs, addressing these concerns would greatly strengthen the manuscript and its contributions to the field.

Reviewer 3:

The authors basically report on the importance of Auger processes in the relaxation and recombination scenario of photoexcited charge carriers. Such processes have been already investigated in InP-based quantum dots as partly mentioned in the text. It is known that strong Auger processes hinder lasing processes.

Our response: It is important to contrast our observation of hot-carrier trapping from Auger losses. Both affect lasing performance of QDs, but the photophysics and potential solutions are different. Seminal work on CdSe-based lasing identified Auger recombination involving band-edge carriers as a complicating factor as it shortens the gain lifetime. This problem was solved over a decade ago with QD designs with charge-carrier delocalization and smooth confinement potentials. Despite these successes with CdSe-based QDs, progress on InP lasing lags behind.

Our work demonstrates what sets InP-based QDs apart from other QD designs: hot-carrier trapping. The losses occur on much faster timescales than Auger recombination. See Fig. 1c for the comparison of timescales. Our work explains how hot-carrier trapping can go unnoticed when optimising InP QDs for display applications. One of the solutions to Auger recombination is charge-carrier delocalisation. In sharp contrast, we propose (newly added during this revision) that to prevent hot-carrier trapping in InP-based QDs one may have to prevent delocalisation across the core-shell interface, which is a likely location of the hot-carrier traps.

Action 13: We have rephrased the discussion paragraph that compares hot-carrier losses in InP-based QDs to the challenge of Auger losses:

“Our results highlight a key challenge in making InP-based QDs ready for lasing application: the suppression of hot-carrier trapping. The challenge for InP-based QDs is clearly different from the challenge faced by CdSe-based QDs two decades ago⁵². The lasing performance of CdSe-based QDs has greatly improved the past two decades mostly owing to core-shell designs that suppress Auger decay. While such steps may also be necessary for InP-based QDs, especially for CW lasing, these may not be sufficient. Indeed, hot-carrier losses reduce gain already before the Auger phase. As the trap emission observed is most consistent with internal traps, the internal structure of the QDs seems the most important target. A potential avenue lies in wave-function engineering by interface polarization⁵³. InP cores can form polarized bonds with the ZnSe shell material that yield inward or outward-pointing dipole moments depending on the InP surface termination. Outward dipole moments could distance the electron wave function from potential interfacial traps on the core-shell interface and prevent hot-carrier trapping. Other strategies could attempt to circumvent the formation of internal and interfacial trap states all together, for example by using MgSe⁵⁴ shelling, which lowers the lattice mismatch between the core and shell material. The difference in hot-carrier trapping probability between the sample of Fig. 2a, 3a and the sample of Fig. 2e, 3d shows a beneficial effect of an oxidation treatment of the core surface during the synthesis^{10,35}.”

(52) Klimov, V. I. *et al.* Optical gain and stimulated emission in nanocrystal quantum dots. *Science* **290**, 314–317 (2000).

Action 14: In the explanation of timescales, we highlight the difference between hot-carrier trapping and Auger recombination with a new sentence:

“Clearly, these losses are not due to simple Auger recombination because they occur on timescales much faster than the biexciton and multiexciton Auger lifetimes (Fig. 1c green).”

Response to reviewer reports on Vonk *et al.* (NCOMMS-24-20930A)

We are happy to read that most of the reviewers' questions from the first round have been addressed to the reviewers' satisfaction. The remaining concerns are important and we are grateful that the reviewers explained them so clearly and constructively. In the new revision round, we have improved our description of the pump–push–probe experiments and analysis, and performed additional analysis to quantify the undesired contribution of multiphoton absorption to our signal. We have also swapped in a different data set in main text Figure 2, which has an even weaker contribution of multiphoton absorption (2.9% relative to hot-carrier trapping) compared to the data set used previously (3.3%).

Below, reviewer comments are reproduced in full in blue, followed by our response, and revisions to the main text and SI reproduced in red.

#####

Reviewer #1 (Remarks to the Author):

The authors have revised the manuscript according to the comments of reviewers and most issues have been well addressed. One additional recommendation would be to keep the Figure Captions more concise as most of the current Figure Captions are much longer than the paragraphs in the main text and a good portion of them are redundant with the main text. Lengthy captions hinder the readers from quickly identifying what is presented in the following figure while reading the paper.

Our response: The figure captions are always a difficult balance between conciseness and completeness.

Action 1: We have taken the reviewer's comment to heart and cut significant parts on *interpretation* from the figure captions in the main text, while making sure that the *descriptions* remain complete. Specifically, our figure captions are now shorter by 20% compared to the previous revision. We do not reproduce the figure captions in this rebuttal document, but refer the reviewer to the revised main text.

#####

Reviewer #2 (Remarks to the Author):

I still have concerns about the pump-push-probe setup based on the authors' reply. They state that: "...the pump–push–probe signal, should not directly contain push–probe contributions such as multiphoton absorption... Furthermore, we note that the first-order effect of multiphoton absorption would be an increased bleach due to the push pulse, because of additional band-edge charge carriers..." This is inaccurate. The differential absorbance, which is the signal comparison between $A_{\text{pump on}}$, push on, probe on and $A_{\text{pump off}}$, push on, probe on measures the effect of the pump pulse. When the push pulse induces multiphoton absorption (including two-photon or three-photon absorption), some carriers are excited to the conduction band. As a result, the difference signal caused by the pump pulse is weakened because fewer carriers are available for excitation. Therefore, the push pulse would lead to a reduced bleach signal, not an increase.

Our response: We are grateful to the reviewer for this clear explanation of the potential effect of multiphoton absorption. Our statement about the first-order effect in the previous rebuttal file was indeed inaccurate.

Action 2: We introduced a new Supplementary Fig. S5, which clearly explains the complete experimental procedure and data analysis behind the pump–push–probe experiments.

Supplementary Fig. S5 | Data-recording procedure for pump-push-probe and pump-probe transients. (a)–(c) Schematic of the data-recording procedure in pump-push-probe experiments. Here, and in what follows, we use a X to indicate blocked pump (XPP), push (XPX), probe (PPX), or any other combination. By placing a chopper in the optical path of the pump laser, we directly measure the differential absorbance [c, $A_{PPP} - A_{XPP}$] between pump on [a, A_{PPP}] and pump off [b, A_{XPP}]. In A_{PPP} a, excitations formed by the pump decrease the absorbance (ground-state absorbance A_0 as reference). 300 ps later, the push instantaneously recovers part of the absorbance by pushing the electron to a hot-carrier state. Because of hot-carrier trapping, part of the initial absorbance bleach is lost after cooling and the absorbance does not reach the absorbance before the push (dashed line). Without the pump A_{XPP} b, the push does (to first order) not interact with the sample and the absorbance is unaffected. The traces in Fig. 2 of the main text are normalised differential absorbance traces such as those in panel c and Fig. S6c. (d)–(f) Same as a–c, but for the pump-probe experiments.

Action 3: We introduced a new Supplementary Fig. S7a, which explains how multiphoton absorption would appear in the transient-absorption data and how it could eventually yield a differential-absorption signal reminiscent of hot-carrier trapping.

Supplementary Fig. S7 | Influence of multiphoton absorption in pump–push–probe experiments.

(a) Multiphoton absorption can look like hot-carrier losses in PPP experiments. In the differential absorbance scheme—assuming no hot-carrier trapping—XPP (middle) induces more absorbance bleach by multiphoton absorption compared to PPP (top), as more QDs are in the ground state upon arrival of the push pulse. This leads to apparent hot-carrier losses in the differential absorbance $A_{\text{PPP}} - A_{\text{XPP}}$ because of a difference in magnitude of multiphoton absorption.

The authors also mention:

“Nevertheless, we characterised the multiphoton absorption of our semiconductor QDs in more detail. We performed pump–push–probe experiments at increasing push fluence and with different push energies. The results are shown below (Supplementary Fig. S6). The apparent hot-carrier losses are constant at low fluence below 10 mJ cm^{-2} but increase for higher push fluences. The increased hot carrier losses for higher fluences must be an indirect effect of interband multiphoton absorption, because the direct effect of interband absorption is the generation (rather than loss) of carriers. Indeed, control experiments with the pump pulse blocked show significant three-photon-induced bleach at push powers exceeding 10 mJ cm^{-2} .”

While the power-dependent experiments provide useful insight into push pulse effects, it should also be noted that push-pulse-induced two-photon absorption follows a quadratic relationship, suggest that the initial start would be very flat. From their push-probe experiments, at 4.6 mJ/cm^2 , multiphoton absorption should already be occurring. Based on their data, I cannot conclude that the observed carrier losses are due to hot carrier trapping.

Our response: The reviewer is correct that “multiphoton absorption should already be occurring” at relatively low push fluences. However, as interband multiphoton absorption is a nonlinear process and intraband absorption is linear, the relative contribution of multiphoton absorption scales with push fluence.

While our previous interpretation of the fluence-dependent data was qualitative, we have now carried out a quantitative analysis of the relative contributions of (nonlinear) multiphoton absorption compared to (linear) hot-carrier losses following intraband absorption. We calculate that the contribution of multiphoton absorption to the apparent hot-carrier losses was 3.3% relative to the real hot-carrier losses, for the data presented in the previous version of the main text.

While the reviewer is correct that multiphoton absorption contributes to our signals, the effect does not pose a challenge to our conclusions. Thanks to this discussion with the reviewer, we now also understand that the increase in apparent hot-carrier losses with push fluence, which we previously interpreted as “an indirect effect”, is actually due to the increasing contribution of multiphoton absorption. We quantify this contribution in the revised Supplementary Fig. S7.

In addition, to avoid any confusion, we swapped the data in Fig. 2 of the main text. By shifting to data recorded with a lower push photon energy, multiphoton absorption turns from a third-order into a fifth-order process. The contribution of multiphoton absorption to the apparent hot-carrier losses is now as low as 2.9% (relative) for the new data in Fig. 2.

We hope that this more complete and quantitative analysis convinces the reviewer that our conclusions on hot-carrier trapping are solid.

Action 4: In Supplementary Note 2 and Supplementary Fig. S7, we now analyse and quantify the contribution of multiphoton absorption to the apparent hot-carrier losses. We fit the apparent hot-

carrier losses to a fluence-independent contribution from real losses and a fluence-dependent contribution from multiphoton absorption:

Influence of multiphoton absorption on pump–push–probe transients.

Supplementary Fig. S7 | Influence of multiphoton absorption in pump–push–probe experiments.

(a) Multiphoton absorption can look like hot-carrier losses in PPP experiments. In the differential absorbance scheme—assuming no hot-carrier trapping—XPP (middle) induces more absorbance bleach by multiphoton absorption compared to PPP (top), as more QDs are in the ground state upon arrival of the push pulse. This leads to apparent hot-carrier losses in the differential absorbance $A_{\text{PPP}} - A_{\text{XPP}}$ because of a difference in magnitude of multiphoton absorption. **(b)** Apparent hot-carrier trapping probability P'_{trap} as a function of push fluence for 0.95 eV (blue) and 0.52 eV (red) push energies. P'_{trap} is independent of push power for small push fluences for both push energies. Solid lines: fits to the data using equation 36, the recovery X from panel **c**, and $n_{\text{low}} = 5$ and $n_{\text{high}} = 3$ for order of the multiphoton absorption. The fits reproduce the nonlinear increase of P'_{trap} with push fluence, showing that multiphoton absorption directly influences the measurement at higher push fluences. **(c)** Fractional bleach loss X as a function of push fluence J . Solid line: fits to data with $X(J) = 1 - \exp(-aJ)$, to account for the saturation of the push-induced bleach loss at $X = 1$. **(d)** Absorbance bleach in a control push–probe experiment with a sub-bandgap push pulse ($E_{\text{push}} = 0.95 \text{ eV}$) and no prior pump pulse on InP/ZnSe/ZnS QDs for varying push fluences. Dashed blue line: absorbance bleach due to the 3.1-eV pump pulse at the fluence used in the PPP experiments of the main text. The absorbance bleach shows a cubic dependence on push fluence (blue solid line) showing that 3-photon absorption populates the band-edge.

Pushing excited charge carriers to hot-carrier states requires high push fluences, which potentially introduces additional band-edge excitations by multiphoton absorption. As such, the differential-bleach recovery X and loss Y may be affected, introducing systematic errors in the apparent hot-carrier trapping probability P'_{trap} . Here, we will discuss the implications of multiphoton absorption in pump–push–probe measurements and how its influence can be minimized and quantified.

In what follows, we use a three-letter notation to indicate a blocked pump (XPP), push (PXP), probe (PPX), or any other combination of light pulses. In our experiments, we measure the differential absorbance, $A_{\text{PPP}} - A_{\text{XPP}}$, using a chopper to correct for push-induced optical artefacts such as the

Stark effect (Fig. S8). Interband multiphoton absorption can introduce apparent hot-carrier losses, as is schematically illustrated in Fig. S7a. Here, XPP induces more absorbance bleach by multiphoton absorption of the push pulse compared to PPP, as more QDs are in the ground state upon arrival of the push pulse. Indeed, this leads to apparent hot-carrier losses in the differential bleach $A_{\text{PPP}} - A_{\text{XPP}}$ —even if all hot carriers cool down with unity efficiency—simply because of a difference in interband multiphoton absorption.

Next, we will derive analytical expressions for the influence of both hot-carrier trapping and multiphoton absorption on the differential absorbance $A_{\text{PPP}} - A_{\text{XPP}}$. The PPP and XPP absorbance bleach directly after arrival of the push pulse ($t = 0$; superscript 0) are given by

$$\Delta A_{\text{PPP}}^0 \propto P_{\text{pump}}(1 - X)$$

$$\Delta A_{\text{XPP}}^0 = 0$$

where P_{pump} is the probability that the pump pulse forms an excitation, and X is the fractional bleach loss, which increases linearly (until saturation; Fig. S7c) with push fluence J . Multiphoton absorption does not contribute to the PPP or XPP bleach at $t = 0$ as the excitations need to cool down before they influence the band-edge absorbance. After cooling (late time; superscript ∞), the PPP and XPP bleach contain contributions of hot-carrier trapping and multiphoton absorption:

$$\Delta A_{\text{PPP}}^\infty \propto P_{\text{pump}}(1 - P_{\text{trap}}X) + (1 - P_{\text{pump}})P_{\text{push}} + P_{\text{pump}}P'_{\text{push}}$$

$$\Delta A_{\text{XPP}}^\infty \propto P_{\text{push}}$$

where a fraction P_{trap} of X is lost because of hot-carrier trapping, P_{push} is the probability of multiphoton excitation by the push pulse of a QD in the ground state, and P'_{push} the probability of multiphoton excitation by the push pulse of a QD already in the excited state. Both P_{push} and P'_{push} increase superlinearly with push fluence J^n , with n (≥ 3 in our experiments) the order of the absorption process. From this, we can compute the apparent trapping probability P'_{trap}

$$P'_{\text{trap}} = \frac{Y}{X} = \frac{\Delta A_{\text{PP}} - (\Delta A_{\text{PPP}}^\infty - \Delta A_{\text{XPP}}^\infty)}{\Delta A_{\text{PP}} - (\Delta A_{\text{PPP}}^0 - \Delta A_{\text{XPP}}^0)} = P_{\text{trap}} + \frac{P_{\text{push}} - P'_{\text{push}}}{X}$$

where $\Delta A_{\text{PP}} = A_{\text{XPP}} - A_{\text{PPP}} = P_{\text{pump}}$. The resulting expression consists of the fluence-independent real trapping probability P_{trap} and a fluence-dependent term, *i.e.* $P'_{\text{trap}} = P_{\text{trap}} + A J^{n-1}$, due to multiphoton absorption. This expression shows that the effect of multiphoton absorption can be minimized by minimizing the push fluence J .

We experimentally validate equation 36 by performing push-fluence-dependent PPP measurements. Fig. S7b shows the apparent trapping probability P'_{trap} as a function of push fluence for low (red, 0.52 eV) and high (blue, 0.95 eV) push energies. For both push energies we observe a constant P'_{trap} at low push fluences and a nonlinear increase at higher push fluences. Next, we determine the fluence-dependent fractional bleach losses X for low (red) and high (blue) push energies (Fig. S7c). We fit the fractional bleach loss to $X(a, J) = 1 - \exp(-aJ)$ to account for saturation of the bleach loss and find the best-fit parameters for low $\hat{a}_{\text{low}} = 0.06 \text{ cm}^2 \text{ mJ}^{-1}$ and high $\hat{a}_{\text{high}} = 0.11 \text{ cm}^2 \text{ mJ}^{-1}$ push energies, respectively. Using these best-fit parameters, we fit the 2 sets of apparent trapping probabilities in Fig. S7b to $P'_{\text{trap}} = P_{\text{trap}} + bJ^n/X(\hat{a}, J)$, with $n_{\text{low}} = 5$ and $n_{\text{high}} = 3$, P_{trap} , and b as fit parameters (Fig. S7b solid lines). There is a good match between data and model, and we conclude that indeed

multiphoton absorption explains the increase of P'_{trap} at higher push fluences. By minimizing the push fluence, we can measure the real trapping probability $P'_{\text{trap}} = P_{\text{trap}}$ from a PPP measurement directly. As an additional control experiment, we independently measure the absorbance bleach in a push–probe experiment (Fig. S7d; 0.95 eV push energy), where we observe a cubic dependence of the absorbance bleach with push fluence. Indeed, the apparent trapping probability P'_{trap} starts to increase (Fig. S7b) from the low-fluence limit P_{trap} , when multiphoton absorption becomes significant (Fig. S7d). The real trapping probabilities P_{trap} are 8.9% for a push photon energy of 0.52 eV and 11.4% for a push photon energy of 0.95 eV. The contribution of hot-carrier trapping to the apparent trapping probability is 97% (relative) for the data in Fig. 2a of the main text.

Action 5: In Fig. 2 of the main text, we now present pump–push–probe data taken with a push photon energy of 0.52 eV:

Fig. 2 | Identifying hot-carrier losses in different QD materials. (a) Differential bleach transient of the band-edge absorbance of our InP/ZnSe/ZnS QDs (PLQY = 95%, $P_{\text{trap}} = 9.2 \pm 0.7\%$) in a pump–push–probe experiment. (I) A 3.1-eV pump laser pulse at $t = -300$ ps generates $\bar{n} = 0.1$ excitations. (II) A 0.52-eV push laser pulse (push fluence of 7.1 mJ cm^{-2}) excites a fraction of the band-edge electrons (Supplementary Fig. S4) to a hot-carrier state, decreasing the absorbance bleach. (III) 91% of the hot charge carriers cool down to the band edge, but 9% of the carriers are lost due to hot-carrier trapping. Solid blue line: fit to an exponential cooling model convolved with the instrument-response function, yielding a time constant of 700 fs. Grey line: deconvolved PPP trace. The y-axis is normalised, setting the bleach just before the push to 1 and immediately after the push to 0. (b–e) Similar PPP experiments on (b) CdSe/CdS/ZnS QDs (PLQY = 95%), (c) CsPbBr₃ nanocrystals (estimated PLQY = 50–90%⁷), (d) CuInS₂/CdS QDs (estimated PLQY = 90%³⁶), and (e) “bare” InP/ZnSe (estimated PLQY = 50%³⁵, $P_{\text{trap}} = 23.6 \pm 0.9\%$), without a ZnS shell.

We mention the potential contribution of multiphoton absorption and refer the reader the Supplementary Fig. S7 for a discussion:

“We minimize the push fluence to minimize the influence of multiphoton absorption (Supplementary Note 2).”

Response to Reviewer comments on NCOMMS-24-20930B, “Hot-Carrier Trapping Preserves High Quantum Yields but Limits Optical Gain in InP-Based Quantum Dots” by Vonk *et al.*

#####

We thank the reviewers for another round of critical reading of our manuscript. The reviewer comments have prompted us to show additional experimental results, making the justification for some of our assumptions more explicit. More specifically, we added measurements on high-quality CdSe-based QDs, where complete conversion from absorption into gain is observed for the 1S transition. Below, we repeat the reviewer comments in blue, provide our point-by-point response, and reproduce any changes to the main text or supplementary information in red.

Reviewer #2 (Remarks to the Author):

The revised manuscript has significantly improved the analysis of the pump-push-probe experiments. The authors conducted push power-dependent experiments to evaluate the relative contributions of hot-carrier losses and multiphoton absorption. They argue that at low push fluences, the influence of multiphoton absorption is minimal, and the constant P_{trap} represents the real hot-carrier trapping probability. However, one critical aspect that the authors have overlooked is the role of carrier recombination during the hot-carrier cooling process. Given that carrier recombination is not particularly slow for these materials, the extracted bleach signal from fitting should not remain constant, as recombination effects would influence the signal. Although the analysis appears reasonable, the data support remains weak, and I still question the manuscript's conclusion that the observed results from pump-push-probe (PPP) experiments are mainly due to hot-carrier losses rather than multiphoton effects. No further review is required; I will leave the decision to the editor.

Our response: We agree with the reviewer that it is important to disentangle effects of hot-carrier trapping and carrier recombination. The reviewer writes that carrier recombination is “not particularly slow for these materials”, but single-exciton recombination is in fact 10^4 times slower than hot-carrier cooling: ~ 10 ns (Fig. 1c) versus 770 fs (Fig. 2a). Multi-exciton recombination is faster, but we prevent contribution of multi-excitons to our experiments by minimizing the pump fluence ($\bar{n} \approx 0.1$) and by choosing a pump–push delay of 300 ps. At 300 ps after the pump pulse, the multi-exciton population is negligible for all pump powers (Fig. 1c).

This final comment by the reviewer might have been triggered by the schematic experiments of Figs. S5 and S7 (added with the previous review round, now Figs. S7,9), which show the time axis not to scale. We updated these figures to avoid confusion, emphasizing the waiting time of 300 ps.

Moreover, to show unambiguously that the pump–push–probe signals are due to hot-carrier losses with negligible contributions of carrier combination, we included new experimental data and updated our analysis to include recombination explicitly.

Action 1: The schematic experiments in Figs. S5a (now Fig. S7a) and S7a (now Fig. S9a) are updated to emphasize the pump–push delay time of 300 ps:

Action 2: We added a decaying background to all fits in Figure 2 and updated the description in Supplementary Note 2.

“We characterise hot-carrier losses using pump–push–probe experiments. In these experiments, we measure the differential absorbance $A_{PPP} - A_{XPP}$ at variable delay times t between the push pulse and the probe (Supplementary Fig. S8a). Such a transient (gray line) is given by

$$|\Delta A(t)| = \Delta A_0 \exp(-k_R t) \{1 - XH(t) + (X - Y)H(t)[1 - \exp(-k_C t)]\}$$

where $H(t)$ is the Heaviside step function, ΔA_0 is the absorbance bleach generated by the pump pulse that is left after the 300-ps waiting time, X is the fractional recovery by excited-state absorption of the electron, Y is the fractional loss after cooling with rate constant k_C , and k_R is the exciton recombination rate. In all our experiments, the exciton recombination rate is much slower than cooling, *i.e.* $k_C \gg k_R$.”

“To extract the differential-bleach recovery X and loss Y from an experiment, we convolve equation 30 with a Gaussian instrument-response function and find

$$|\Delta A(t)| = \Delta A_0 \exp(-k_R t) \left(1 - \frac{X}{2} \left[1 + \operatorname{erf} \left(\frac{t}{\sqrt{2}\delta} \right) \right] + \frac{X - Y}{2} \left\{ 1 + \operatorname{erf} \left(\frac{t}{\sqrt{2}\delta} \right) - \exp \left[\frac{k_C}{2} (k\delta^2 - 2t) \right] \left[1 - \operatorname{erf} \left(\frac{k_C \delta^2 - t}{\sqrt{2}\delta} \right) \right] \right\} \right)$$

where δ is the width of the instrument-response function, and $\text{erf}(x)$ is the error function.”

Action 3: We added an experiment to Fig. 2a of the main text, directly comparing transient-absorption experiments with and without push pulse. We also extended the range of push–probe delay times to demonstrate the slow dynamics of recombination. As carrier recombination occurs in both experiments, the clear difference in bleach level after hot-carrier cooling (5–300 ps) must be due to losses during the cooling process.

“Differential bleach transient of the band-edge absorbance of our InP/ZnSe/ZnS QDs (PLQY = 95%) in a pump–push–probe experiment (red; $P_{\text{trap}} = 7.5 \pm 0.8\%$ at 0.52 eV push energy) and in a reference experiment without push (gray). Red line: fit to an exponential cooling model convolved with the instrument-response function on top of experimental decay due to carrier recombination. Black line: only exponential decay due to carrier recombination. The fitted time constants are 770 fs for cooling and 8 ns for recombination.”

Reviewer #4 (Remarks to the Author):

I understand from reading the material sent that my part was to decide between two conflicting views of the potential artifacts which might arise from multi-photon absorbance of the Push pulse on the PPP data. In order to calibrate my response I first read the entire article and find that significant misconceptions have been incorporated into the analysis which make publication of this paper premature.

The assumption that absence of a full reversal of band edge absorption to stimulated emission after absorbing multiple photons is due to trapping goes against prevailing opinion in the field. Starting with the earliest power dependent studies of CdSe nano-crystals by Klimov and Co, the induced bleach saturated at sample transparency, with no sign of stimulated emission. Furthermore a single exciton was shown to induce a 50% reduction in the lowest exciton absorbance. Ironically, the mechanism behind this is still debated, but is often assigned to degeneracy of the hole states which reduce their contributions to state filling. Since the authors assume similarity in electronic states between CdSe and InP, it stands to reason that a similar explanation applies here as well. The above assumption is not a limited issue and is at the heart of all analysis here. Therefore not dealing with this issue is a major shortcoming which requires further thought.

Our response: This is an important and interesting alternative explanation offered by the reviewer. We should have addressed this in the earlier versions of our manuscript. In the revised manuscript we make up for this with new experimental data and analysis. Specifically, we now (1) present bleach data on high-PLQY CdSe-based QDs, showing that they follow the state-

filling model with the theoretical hole degeneracy of $g_h = 4$; (2) attempt and fail at fitting the InP bleach data to a model with increased g_h .

Moreover, we revised the abstract of our manuscript and our discussion surrounding Fig. 1 to more clearly convey what makes InP-based QDs special. As the reviewer states correctly, earlier generations of CdSe-based QDs may have struggled to support high gain. The key difference is what happened as the material quality improved. For CdSe, the gain performance improved as the PLQY improved. For InP, in contrast, the gain performance is still poor although the PLQY is near 100%. This contrast in gain performance is visually illustrated with the new data in Fig. 1.

Action 4: We added a new plot to Fig. 1 (panel f), showing transient-absorbance spectra of CdSe/CdS/ZnS QDs at increasing pump fluence.

“(f) Comparison to CdSe-based QDs: transient-absorbance spectra of CdSe/CdS/ZnS QDs after cooling ($t = 1$ ps) as a function of excitation number \bar{n} . Data reused from Ref. 29.”

[29] Geuchies, J. J. *et al.* Quantitative electrochemical control over optical gain in quantum-dot solids. *ACS Nano* **15**, 377–386 (2021).

Action 5: In Fig. 1e, we added the bleach data of the CdSe/CdS/ZnS QDs. The data points match the state-filling model with $g_h = 4$ and $g_e = 2$.

“(e) The maximum bleach for InP/ZnSe/ZnS QDs (blue; $t = 1$ ps) and CdSe/CdS/ZnS QDs (black; $t = 1$ ps) as a function of the excitation number \bar{n} . See Supplementary Note 1 for our calculation of \bar{n} . Black line: prediction of a state-filling model for the maximum bleach, using electron and hole band-edge degeneracies $g_e = 2$ and $g_h = 4$. Arrow: the experiment on InP-

based QDs at $\bar{n} = 7.1$ produces an initial bleach consistent with an effective number of excitations of no more than $\bar{n}_{\text{eff}} = 1.5$.”

Action 6: The abstract now more clearly conveys the main motivation behind our study, which is the striking contrast between poor gain properties of InP-based QDs and their excellent quantum yield under low illumination:

“Here, we report unusual photophysics of state-of-the-art InP-based quantum dots, which makes them unattractive as a laser gain material despite a near-unity quantum yield.”

“This process reduces the achievable population inversion and limits light amplification for lasing applications. However, it does not quench fluorescence. Instead, trapped carriers can recombine radiatively, leading to delayed—but bright—fluorescence. Single-quantum-dot experiments confirm the direct link between hot-carrier trapping and delayed fluorescence. Hot-carrier trapping thus explains why the latest generation of InP-based quantum dots struggle to support optical gain, although the quantum yield is near unity for low-intensity applications.”

Action 7: In the main text, we now (1) describe the striking difference between CdSe- and InP-based QDs more clearly; and (2) briefly consider the model of higher hole degeneracy, referring the reader to Supplementary Information Fig. S2 for our fitting attempt:

“The maximum bleach we observe in the experiment is weaker than expected (Fig. 1d). The band-edge absorbance bleach of our InP-based QDs saturates around $A \rightarrow 0$ as the excitation fluence is increased, while complete population inversion would lead to $A \rightarrow -A_0$ (where A_0 is the ground-state absorbance; Fig. 1e). Limited absorbance bleach in QDs was previously explained in terms of a high degeneracy of states at the valence-band edge²⁴ but this explanation does not reproduce the observed gain saturation at $A \rightarrow 0$ (Supplementary Information Fig. S2)²¹. Instead, our data is consistent with the interception of photogenerated charge carriers on the timescale of cooling, a process commonly referred to as “hot-carrier trapping”. These carriers never reach the band edge and do not contribute to the band-edge absorbance bleach nor gain. Historically, this problem occurred in CdSe-based QDs, but charge-carrier trapping could be prevented with improvements to the QDs’ surface passivation, simultaneously leading to higher PLQYs of >80% and improved gain (Fig. 1f). These successes were consistent with the existing paradigm that PLQY and charge-carrier trapping are intimately linked. In contrast, in our InP-based QDs, the PLQY is high (95%) but the gain still appears limited by hot-carrier trapping. Below, we will obtain more direct evidence for hot-carrier trapping and find out why the PLQY is, nevertheless, near unity.”

[24] Klimov, V. I. Optical nonlinearities and ultrafast carrier dynamics in semiconductor nanocrystals. *J. Phys. Chem. B* **104**, 6112–6123 (2000).

[21] Sousa Velosa, F. *et al.* State filling and stimulated emission by colloidal InP/ZnSe core/shell quantum dots. *Adv. Opt. Mater.* **10**, 202200328 (2022).

Action 8: We added Fig. S2 to the revised Supplementary Information, showing that a model with increased hole degeneracy does not adequately produce a saturation of the bleach, but instead predicts complete inversion of the absorption at higher excitation numbers.

“Supplementary Fig. S2 | Maximum bleach for different hole degeneracies. Maximum bleach as a function of excitation number \bar{n} for different hole degeneracies between $g_h = 2$ –10 (fixed electron degeneracy $g_e = 2$). All hole degeneracies show complete population inversion at high excitation powers, showing that the experimental results on InP/ZnSe/ZnS cannot be explained by invoking a high hole degeneracy.”

Some additional points that I suggest the authors address are:

The extensively normalized format of the PPP presentation is problematic. What is the fraction of the bleach which is erased due to the push and why not present it as such?

Our response: We considered various ways of presenting the data. We agree with the reviewer that it is important that readers can appreciate the absolute numbers, if they wish so. At the same time, the presentation in Fig. 2 should emphasize the main message, which is that the *fractional* recovery is different between samples.

Nevertheless, we want to provide absolute numbers to the reader. In the previous version of the manuscript, Supplementary Fig. S6 showed the absolute absorbance bleach as a function of time for the experiment in Fig. 2a. A new Supplementary Fig. S5 presents such non-normalised data for all other experiments in Fig. 2b–e. Moreover, in the revised manuscript, we added the fraction of bleach erased by the push for each experiment to the caption of Fig. 2.

Action 9: The caption of Fig. 2 lists the fraction of bleach erased by the push for each experiment:

“The fractional decrease of the absorbance bleach due to the push pulse is 37%, 49%, 62%, 19% and 40% for the experiments in Fig. 2a–e, respectively. See also Supplementary Fig. S11.”

Action 10: A newly added Supplementary Fig. S5 shows non-normalized pump–push–probe data for all other experiments in Fig. 2b–e and over a wider range of push–probe delay times:

“Supplementary Fig. S11 | Non-normalised pump–push–probe experiments. (a) Non-normalised pump–push–probe experiment on CdSe/CdS/ZnS QDs, showing a fractional bleach recovery $X = 49\%$. **(b)–(d)** Same as **a**, but for **b** CsPbBr₃ perovskite nanocrystals ($X = 62\%$), **c** CuInS₂/CdS ($X = 19\%$), and **d** InP/ZnSe ($X = 40\%$). Bleach is averaged over the band-edge absorption band at 1.90–2.10 eV, 2.42–2.46 eV, 2.00–2.10 eV, and 2.12–2.15 eV for panels **a–d**, respectively.”

Why is the probe delay limited to 4 ps, and why waste half of each PPP frame on negative times which teach us nothing? Seeing that the cooling time for InP seems slowest, wouldn't the authors do well to extend the delay to verify cooling is complete?

Our response: The bleach at negative time delays serves as the reference point. We wanted to demonstrate that it is constant over a few ps, because this means that no carrier recombination occurs on the time scale of cooling. See the comment of Reviewer 2 above.

Inspired by this comment and that of Reviewer 2, we extended the time axis of our pump–push–probe experiments on the InP/ZnSe/ZnS QDs (Fig. 2a) from -5 ps to 300 ps. This demonstrates that cooling is complete within 5 ps. See **Action 2**. We also extended the time range on the experiments of the other samples in a new Supplementary Fig. S5. See **Action 10**.

Is it a coincidence that the remainder of non-restored bleach is only observed in the most noisy PPP data, which probably indicates very weak push effects?

Our response: Based on reviewer comments in previous rounds, we thoroughly investigated the effect of push fluence. The data is presented in Supplementary Fig. S9 (previously S7). This shows clearly that non-restored bleach occurs at all push fluences and, in fact, becomes more significant with increasing push fluence. By presenting data with relatively low push fluences in the main text, we avoid any potential contribution of multiphoton absorption to the non-restored bleach. This comes at the cost of some increased noise on the data.

Action 11: To quantify the effect of noise on the determination of hot-carrier trapping probabilities, we propagate the error on the fit to the bleach dynamics. The error estimates are included in the caption of Fig. 2. They are all below 1%. In the main text we write:

“This means that $P_{\text{trap}} = 7.5 \pm 0.8\%$ (error provided as one standard deviation) of the hot electrons are lost (Fig. 2a, inset III).”

The errors on the trapping probabilities are also presented as error bars in the updated Supplementary Fig. S9b (previously Fig. S7).

“Error bars represent one standard error and are propagated from the fit to the cooling dynamics.”

The PPP data is presented as a single channel signal, but the experiment should provide a broad band difference spectrum. Have the authors followed the push induced delta a during the cooling? Does it resemble the spectral change observed for cooling following simple PP exciting in the blue? Is the PPP signal the result of integrating over a range of wavelengths, or truly a single channel result? Not learning from the PPP spectrum seems strange.

Our response: We indeed measure a broad-band difference spectrum in our PPP experiments. For the bleach values and time traces in the main text (Fig. 1c,e–g and Fig. 2), we have averaged over a range of photon energies around the band-edge bleach between 565 and 600 nm. It is important to mention these numbers in the manuscript.

Sparked by the Reviewers’ interest, we have further investigated the difference spectrum during cooling after the pump pulse and push pulse, respectively. The spectra are presented in a new Supplementary Information Figure S6. The spectral similarity between cooling following the push pulse (infrared) and later stages of cooling following a pump pulse (blue) confirms that our PPP experiments track electron cooling, which is known to be the rate-limiting step in hot-carrier cooling.

To summarize, the bleach spectra grow in more or less evenly over the full spectral range during 2 ps, following either pump (blue) or push (infrared) excitation (Fig. S6c,d). This indicates that, in both experiments, the cooling dynamics are limited by electron cooling over the relatively large $1P_e-1S_e$ gap as the slowest process. This produces a broad spectral response, because transitions involving the $1S_e$ electron are possible both in the core (2.0–2.4 eV) and the shell (>2.4 eV). For the pump experiments, the first few measurements (<400 fs) show a different spectral shape. This is most apparent from the normalised spectra of Fig. S6e,f. The distinctive early spectra are consistent with the presence of hot electrons and hot holes in the first 400 fs following interband excitation. While holes cool down within 400 fs, the final step in electron cooling takes a further few hundred fs to completion.

Action 12: We now mention the spectral range over which we average the bleach of InP/ZnSe/ZnS QDs in the caption of Fig. 2a. The spectral ranges used for the other samples are listed in Supplementary Fig. S11. See **Action 9**.

Action 13: In the revised Supplementary Information we present the ΔA spectra following pump and push excitation:

“Supplementary Fig. S6 | Bleach spectrum during cooling after pump or push. (a) The absorbance bleach averaged over 565–600 nm as a function of delay time with respect to pump excitation at 3.1 eV. The ingrowth reveals carrier cooling. **(b)** Same, but with respect to the push excitation at 0.52 eV. **(c)** Bleach spectra for three selected delay times following the pump pulse, as indicated in panel **a**. **(d)** Same, but following the push pulse. The dashed line is the bleach spectrum before the push. We observe that the bleach grows in evenly over the full spectral range in both panels **c** and **d**. In panel **d**, the bleach does not recover completely, which we ascribe to hot-electron losses. **(e,f)** Normalized bleach spectra at the same delay times as in panels **c,d**, respectively. These show more clearly that the bleach grows in evenly, except for the earliest delay time following the pump (panel **e**). The earliest spectrum after the pump is consistent with the presence of hot holes and hot electrons in levels higher than $1P_e$. At later delay times, the spectra following pump (panel **e**) and push (panel **f**) excitation are similar. This indicates that, in both experiments, the cooling dynamics are limited by electron cooling over the relatively large $1P_e-1S_e$ gap as the slowest process, with a time constant of a few hundred fs.^{S7,S8”}

[S7] Hendry, E.; Koeberg, M.; Wang, F.; Zhang, H.; de Mello Donega, C.; Vanmaekelbergh, D.; Bonn, M. Direct observation of electron-to-hole energy transfer in CdSe quantum dots *Phys. Rev. Lett.* 2006, **96**, 057408

[S8] Prins, P.T.; Spruijt, D.A.W.; Mangnus, M.J.J.; Rabouw, F.T.; Vanmaekelbergh, D.; de Mello Donega, C.; Geiregat, P. Slow hole localization and fast electron cooling in Cu-doped InP/ZnSe quantum dots *J. Phys. Chem. Lett.* 2022, **13**, 9950–9956.

Action 14: In the revised main text, we refer to the new Supplementary Fig. S6 and mention that the absorbance changes are consistent with electron $1S \rightarrow 1P$ excitation:

“The absorbance changes induced by the push pulse are consistent with excitation of conduction-band electrons from the 1S to the 1P level (Supplementary Fig. S5, S6). In our experiments, a probe pulse measures the differential absorbance (difference between pump+push and push-only, see Methods and Fig. S7 for details) at variable time delays with respect to the push. We minimize the push fluence to minimize the influence of multiphoton absorption (Supplementary Note 2). Exciting charge carriers by the push pulse lowers the band-edge bleach, which is restored as excited charge carriers cool down over a timescale of 770 fs. Importantly, only 92% of the initial bleach is recovered (Supplementary Note 2, corrected for the instrument-response function).”

For all the above reasons I find this paper not suitable for publication at this time.

Our response: We hope that our new data on CdSe-based QDs, the added model of higher degeneracy, and the improved presentation of some of the pump–push–probe data have convinced the Reviewer.

Response to Reviewer comments on NCOMMS-24-20930C, “Hot-Carrier Trapping Preserves High Quantum Yields but Limits Optical Gain in InP-Based Quantum Dots” by Vonk *et al.*

#####

We thank Reviewer #4 for another critical look at our manuscript. It is encouraging that the Reviewer appreciates our efforts to improve the manuscript and now recommends that the paper is ready for publication. We have considered the final remarks of the reviewer and made changes in response, as detailed below.

Reviewer #4 (Remarks to the Author):

It is obvious that the authors have seriously considered the criticism of all reviewers, including my own, and as a result the paper is more balanced. None the less I am not convinced by the authors arguments for the following reasons. I find the perfect match between the levels of signal saturation, full transparency before Auger recombination, and a reduction to 50% bleach - exactly as reported decades ago for CdSe cores, just too much of a coincidence to arise from a totally different mechanism here. I note that work of colleagues who are authors here has also shown time and again that carrier cooling in virtually all semiconductor quantum dots leads not only a buildup of the lowest exciton bleach, but also to significant changes in the pump-probe spectrum, including disappearance of state filling bleaches higher up, and appearances of both net absorption as well as bleaches, and in particular an induced absorption below the band gap. I therefore wonder how they accept that the push is exciting the electrons to higher levels but for some reason causes no similar effects on the bleach spectrum during the cooling which follows. I would therefore recommend that the authors at least comment on these unusual points, but leave that option to their discretion. Thus I find the paper acceptable for publication as is, with the recommendation above. I rest my case.

Our response: We thank the Reviewer for the questions that prompted us to contrast the behavior of state-of-the-art InP and state-of-the-art CdSe QDs more clearly. It is indeed quite a mystery that, as the quantum yield of both materials increased over the past decades, InP QDs still show gain properties similar to CdSe decades ago. Our work investigates the bleach evolution (Fig. 1) further with pump–push–probe (Fig. 2), long-timescale photoluminescence (Fig. 3), and single-QD spectroscopy (Fig. 4). The combination provides compelling evidence for hot-electron trapping and slow release.

The Reviewer is correct that the bleach spectra directly following excitation—with either pump or push—are not yet fully understood. In fact, even the linear absorption spectrum of InP QDs is a topic of discussion, about which some of us have just published last week.^{R1} This combined experimental and theoretical work showed that the $1P_e$ excited conduction-band level at higher energies is not distinguishable in the linear absorption spectrum. In the revised manuscript, we discuss our ultrafast transient-absorption spectrum in more detail in this context.

[R1] D. Respekta, P. Schiettecatte, L. Giordano, N. De Vlamynck, P. Geiregat, J.I. Climente, Z. Hens. Energy-Level Structure and Band Alignment in InP/ZnSe Core/Shell Quantum Dots, *ACS Nano*, in press (2025). doi.org/10.1021/acsnano.5c02258

Action: We have followed the Reviewer’s recommendation. The caption of Supplementary Fig. S6 now describes the transient absorption spectra during the first ps after excitation in more detail, highlighting absence of clear state-filling bleaches at higher energy:

“(e,f) Normalized bleach spectra at the same delay times as in panels c,d, respectively. These show that the bleach grows in evenly over the spectral range between 2.0 eV and 2.5 eV. Following the assignments of Ref. S7, we assign the bleach peak at 2.15 eV to the $1S_h-1S_e$ transition and the one at 2.4 eV to the $2S_h-1S_e$ transition. Their even recovery is thus consistent with excitation of the electron, which affects the two transitions equally. We do not observe transient bleach signatures at higher energy due to filling of the $1P_e$ electron level. This unexpected absence of clear $1P_e$ signals indicate that the $1P_e$ energy falls within the ZnSe shell conduction band.^{S7} Hence, the high degeneracy of shell-delocalized states and reduced overlap with hole states reduces the contribution of the $1P_e$ level to (transient) absorption spectra. The overlap of different valence-band levels with the $1S_e$ level, on the other hand, is good. This, combined with the significant overall broadening of spectral features of InP QDs, explain why bleach spectra change evenly over a broad spectral range during cooling and no clear $1P_e$ signals are observed. The similarity of the spectra following pump (panel e) and push (panel f) excitation indicate that, in both experiments, the cooling dynamics are limited by electron cooling over the relatively large $1P_e-1S_e$ gap as the slowest process, with a time constant of a few hundred fs.”

[S7] D. Respekta, P. Schiettecatte, L. Giordano, N. De Vlamynck, P. Geiregat, J.I. Climente, Z. Hens. Energy-Level Structure and Band Alignment in InP/ZnSe Core/Shell Quantum Dots, *ACS Nano*, in press (2025). doi.org/10.1021/acsnano.5c02258